# Smac mimetics synergize with immune checkpoint inhibitors to promote tumour immunity against glioblastoma

Shawn T. Beug[1,2], Caroline E. Beauregard[1,2], Cristin Healy[1,2], Tarun Sanda[1,2], Martine St-Jean[1], Janelle Chabot[1], Danielle E. Walker[1], Aditya Mohan[1], Nathalie Earl[1], Xueqing Lun[3], Donna L. Senger[3], Stephen M. Robbins[3], Peter Staeheli[4], Peter A. Forsyth[5], Tommy Alain[1,2], Eric C. LaCasse[1,*] & Robert G. Korneluk[1,2,*]

Small-molecule inhibitor of apoptosis (IAP) antagonists, called Smac mimetic compounds (SMCs), sensitize tumours to TNF-α-induced killing while simultaneously blocking TNF-α growth-promoting activities. SMCs also regulate several immunomodulatory properties within immune cells. We report that SMCs synergize with innate immune stimulants and immune checkpoint inhibitor biologics to produce durable cures in mouse models of glioblastoma in which single agent therapy is ineffective. The complementation of activities between these classes of therapeutics is dependent on cytotoxic T-cell activity and is associated with a reduction in immunosuppressive T-cells. Notably, the synergistic effect is dependent on type I IFN and TNF-α signalling. Furthermore, our results implicate an important role for TNF-α-producing cytotoxic T-cells in mediating the anti-cancer effects of immune checkpoint inhibitors when combined with SMCs. Overall, this combinatorial approach could be highly effective in clinical application as it allows for cooperative and complimentary mechanisms in the immune cell-mediated death of cancer cells.

[1] Apoptosis Research Centre, Children's Hospital of Eastern Ontario Research Institute, 401 Smyth Road, Ottawa, Ontario, Canada K1H 8L1. [2] Department of Biochemistry, Microbiology and Immunology, University of Ottawa, 451 Smyth Road, Ottawa, Ontario, Canada K1H 8M5. [3] Arnie Charbonneau Cancer Institute, University of Calgary, 2500 University Drive NW, Calgary, Alberta, Canada T2N 1N4. [4] Institute of Virology, University Medical Center Freiburg, Freiburg D-79104, Germany. [5] H. Lee Moffit Cancer Center and Research Institute, 12902 Magnolia Drive, Tampa, Florida 33612, USA. * These authors contributed equally to this work. Correspondence and requests for materials should be addressed to E.C.L. (email: eric@arc.cheo.ca) or to R.G.K. (email: bob@arc.cheo.ca).

Evasion of apoptosis and avoidance of immune attack represent two key hallmarks of cancer[1]. Members of the inhibitor of apoptosis (IAP) gene family play important interconnecting roles in both of these characteristic pathways of tumorigenesis[2], providing a critical nexus in the targeting of cancer. Small-molecule antagonists of the IAPs, known as Smac mimetic compounds (SMCs), are in clinical development for cancer therapy[3]. SMCs were found to exert immunological effects leading to the eradication of tumours[4,5]. Mechanistically, SMCs bind to cellular IAP 1 and 2 (cIAP1 and cIAP2), which induces the auto-ubiquitination and subsequent proteasomal-mediated degradation of these IAPs[6]. At higher doses, SMCs can antagonize X-linked IAP (XIAP), de-repressing the ability of XIAP to inhibit pro-apoptotic caspases. These three IAPs are E3 ubiquitin ligases that control diverse signalling pathways through post-translational ubiquitination reactions, including pathways central to immunity[7]. The SMC drug-sustained loss of IAPs has important consequences. First, SMC-mediated antagonism of the IAPs sensitizes cancer cells to death ligands originating from the immune system by switching tumour necrosis factor alpha (TNF-α) from a survival factor to a potent death factor, leading to death through the ripoptosome or the necrosome[8,9]. Second, the loss of the cIAPs activates the alternative nuclear factor kappa B (NF-κB) pathway through the stabilization of NF-κB-inducing kinase (NIK) in cells[10]. NIK is a target of the cIAPs, wherein NIK is constitutively ubiquitinated and degraded. However, on binding of a TNF superfamily ligand to their cognate receptor, such as CD137 (aka, 4-1BB), the cIAPs are sequestered and degraded, thereby allowing for the accumulation of NIK and activation of the alternative NF-κB pathway[10,11].

In general, tumours are resistant to the induction of apoptosis due to the p53-mediated adaptations of the intrinsic mitochondrial cell death pathway to damaging DNA lesions and prior chemotherapy treatments[12]. In contrast, the extrinsic cell death pathway, which responds to death ligands from the immune system, is typically intact in cancer cells[12]. Thus, the extrinsic pathway provides an avenue to exploit for the induction of cancer cell death. However, tumours have evolved other means to suppress immune attack such as by upregulating T-cell co-inhibitory molecules, typified by Programmed death-ligand (PD-L1, a.k.a., CD274), on the cancer cell surface. The recent clinical successes for antibody-based biologics, called immune checkpoint inhibitors (ICIs), which target molecules like programmed cell death protein 1 (PD-1, a.k.a. CD279), have demonstrated remarkable efficacy[13–15]. ICIs overcome the countervailing immune checkpoint blockade and promote the immune system to attack tumour cells. However, these drugs are not without limitations: a notable example is the appearance of limiting toxicities related to the induction of autoimmunity.

Here, we investigate the efficacy of targeting cIAP1 and cIAP2 with a SMC in combination with an immunotherapy agent for the treatment of glioblastoma. We demonstrate that SMCs and ICIs combine to form an effective immunotherapy for the treatment in mouse models of this deadly brain cancer, and for other cancers, such as mammary carcinoma and multiple myeloma/plasmacytoma. In addition to the synergy that we have found with innate immune stimulants, our results uncover a second important mechanism by which SMCs exert their anti-cancer effects, specifically through the potentiation of cytotoxic T-cell (CTL) activity against tumours, which is amplified with an ICI.

## Results

### Combining immunostimulatory agents for glioblastoma therapy.
We previously found that SMC-mediated death of cancer cell lines was potentiated with a type I IFN-inducing oncolytic virus, such as the attenuated rhabdovirus Vesicular stomatitis virus

(VSVΔ51)[16]. VSVΔ51 infection lacks cytolytic activity for the tested glioma cell lines, presumably due the presence of a partial type I interferon (IFN) response[17]. Notably, cIAP2 (a mediator and an indicator of NF-κB activity) is upregulated in glioblastoma tumours, showing that cIAP2 promotes tumorigenesis and further drives therapeutic resistance[18–20]. We show here that cultured and primary glioblastoma cell lines are killed with SMC when combined with exogenous TNF-α, the oncolytic virus VSVΔ51 or with an infectious but non-replicating virus, VSVΔ51ΔG (Fig. 1a,b). We confirmed that the synergistic effects between the SMC, LCL161, and TNF-α is a general phenomena within this drug class, as we observed death of glioblastoma cells with the combination of TNF-α and different SMCs (Supplementary Fig. 1). Furthermore, we also observed potentiation of SMC efficacy with the oncolytic rhabdoviruses, VSVΔ51 or Maraba-MG1, for human brain tumour-initiating cells (BTICs) (Fig. 1c). Non-replicating rhabdovirus particles (NRRPs), which retain their infectious and immunostimulatory properties without the ability to replicate[21], similarly were found to synergize with SMCs to induce glioblastoma cell death (Supplementary Fig. 2). Notably, only ∼50% of profiled cancer cell lines are sensitized to death in combination of SMC and TNF-α or TNF-related apoptosis-inducing ligand (TRAIL); the majority of resistant cell lines are further sensitized to death with the downregulation of the caspase-8 inhibitor, cFLIP (cellular FLICE-like inhibitory protein)[22]. Consistent with this previous finding, two glioblastoma lines that are refractory to combined treatment with SMC and TNF-α or VSVΔ51ΔG were killed upon silencing of cFLIP (Supplementary Fig. 3). Normal diploid human fibroblasts, in contrast, were not sensitized to cell death with the downregulation of cFLIP and combined treatment. These findings suggest that an IFN and/or cytokine response, and not direct virus-induced cytolysis, are responsible for the SMC-induced death of glioblastoma cells.

Since VSVΔ51 is neurotoxic, and since issues remain about the 'immune privileged' brain microenvironment and penetration of drugs across the blood–brain barrier (BBB), we set out to test the effects of systemic and intracranial immunotherapy agent delivery. Following the establishment of intracranial CT-2A tumours (Supplementary Fig. 4), we tested whether systemic administration by oral gavage of the SMC, LCL161, could cause the transient degradation of its primary targets proteins, cIAP1 and cIAP2, within intracranial murine tumours. We observed downregulation of cIAP1/2 within CT-2A brain tumours in SMC-treated mice (Fig. 2a). In contrast, we did not observe downregulation of the cIAPs in neighbouring non-tumorous brain tissue (Fig. 2a) nor in the cortex or cerebellum in non-tumour bearing mice (Supplementary Fig. 5). Therefore, SMCs have the capacity to reach tumours within the brain that have a compromised BBB. The systemic administration of immunostimulatory agents, such as the synthetic TLR3 agonist poly(I:C) injected intraperitoneally (i.p.) or the oncolytic virus VSVΔ51 administered intravenously (i.v.), induced the production of cytokine TNF-α in the serum and brain of non-tumour bearing mice (Fig. 2b). Our results are in agreement with a report demonstrating that systemic infection or immune stimulation induces cytokine production including type I IFN within the brain[23]. When mice bearing intracranial CT-2A glioblastoma were treated singly with SMC (oral gavage), VSVΔ51 (i.v.) or poly(I:C) (intracranially, i.c.), the extension of mouse survival was minimal for this aggressive cancer (17% survival rate) (Fig. 2c). However, the combination of systemic SMC with an immunostimulatory trigger, VSVΔ51 or poly(I:C), significantly extended survival and resulted in durable cures for 71 or 86% of the mice, respectively. Tumours (which were not tagged with a foreign protein to avoid enhanced immunity) were imaged at day 40 post-implantation by MRI to confirm the observed treatment outcomes (Fig. 2d).

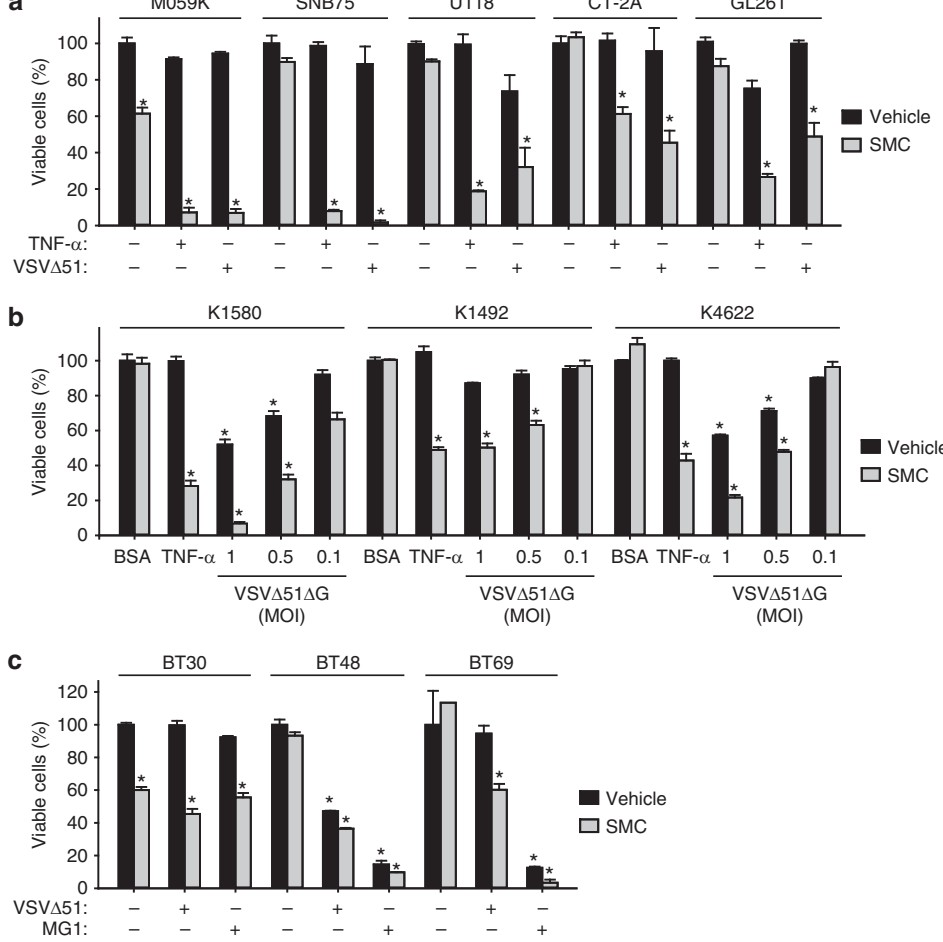

**Figure 1 | SMCs induces the death of glioblastoma cells in the presence of cytokines or oncolytic viruses.** (**a**) Alamar blue viability assay of human (M059K, SNB75, U118) and mouse (CT-2A, GL261) glioblastoma cells treated with vehicle or 5 μM LCL161 (SMC) and 0.1 ng ml$^{-1}$ of TNF-α or 0.01 MOI of VSVΔ51 for 48 h. Error bars, mean, s.d. $n = 4$. (**b**) The indicated primary mouse NF1$^{-/+}$ p53$^{-/+}$ lines were treated with vehicle or 5 μM LCL161 and 0.01% BSA, 1 ng ml$^{-1}$ TNF-α or the indicated MOI of a nonspreading version of VSVΔ51 (VSVΔ51ΔG) for 48 h, and viability was assessed by Alamar blue. Error bars, mean, s.d. $n = 4$. (**c**) Alamar blue viability assays of human brain tumour-initiating cells (BTICs) treated with vehicle or 5 μM LCL161 and 0.001 MOI of VSVΔ51 or Maraba-MG1 for 48 h. Error bars, mean, s.d. $n = 3$. (**a,b**) Representative data from three independent experiments using biological replicates. Statistical significance was compared with vehicle and BSA treatment using ANOVA using Dunnett's multiple comparison test. Significance is reported if $P < 0.0001$ (*).

The virus-induced immune effects are mediated in part by type I IFNs. We show here that CT-2A cells are partially sensitive to combined SMC and recombinant IFN-α *in vitro* (Fig. 2e). We observed that the intracranial administration of SMC resulted in even more profound degradation of the IAP proteins in CT-2A brain tumours (Supplementary Fig. 6). For *in vivo* studies, we used a form of recombinant IFN-α that consists of a hybrid of human isoforms IFN-α B and IFN-α D, which displays potent antiviral activity among a broad range of species[24]. A single co-administration of SMC and IFN-α significantly extended mouse survival and resulted in a 50% durable cure rate (Fig. 2f). Long-term survivors displayed no overt physical or behavioural defects from the single or combined intracranial treatments of SMC, poly(I:C) or IFN-α (Fig. 2c,f; Supplementary Fig. 7). Furthermore, as we observed a transient increase of intracranial TNF-α within the brain on systemic VSVΔ51 infection or treatment with poly(I:C) (Fig. 2b), we sought to determine whether systemic administration of recombinant IFN-α alongside with SMC treatment would be efficacious in the CT-2A glioblastoma model. Similar to the combination of SMC and VSVΔ51, the combination of IFN-α administered i.p. with oral gavage of SMC resulted in durable cures in 55% of the

mice (Fig. 2g). These results suggest that the presence of a transient inflammatory environment in the brain is tolerable and indicate that indirect and other direct (intracranial) routes of combination treatment administration may be feasible.

**Generation of long-term tumour immunity in cured mice.** The innate immune system is a key player in the SMC-mediated death of tumour cells[16]. Nevertheless, fundamental questions remain as to the contributory role of the adaptive immune system in this SMC combination approach. Furthermore, a potential pitfall of the proposed use of oncolytic viruses or other immunostimulatory agents in combination with SMC treatment could be the increase in expression of checkpoint inhibitor ligands on cancer cells, thereby negating CTL-mediated attack of tumours[25]. Flow cytometry analysis revealed that treatment of glioma cells with recombinant type I IFN or infection with VSVΔ51, but not treatment with TNF-α, resulted in the increased surface expression of PD-L1 and major histocompatibility complex (MHC) I markers. Moreover, there was no significant impact on the expression of these tumour surface molecules by SMC treatment (Fig. 3a; Supplementary Fig. 8).

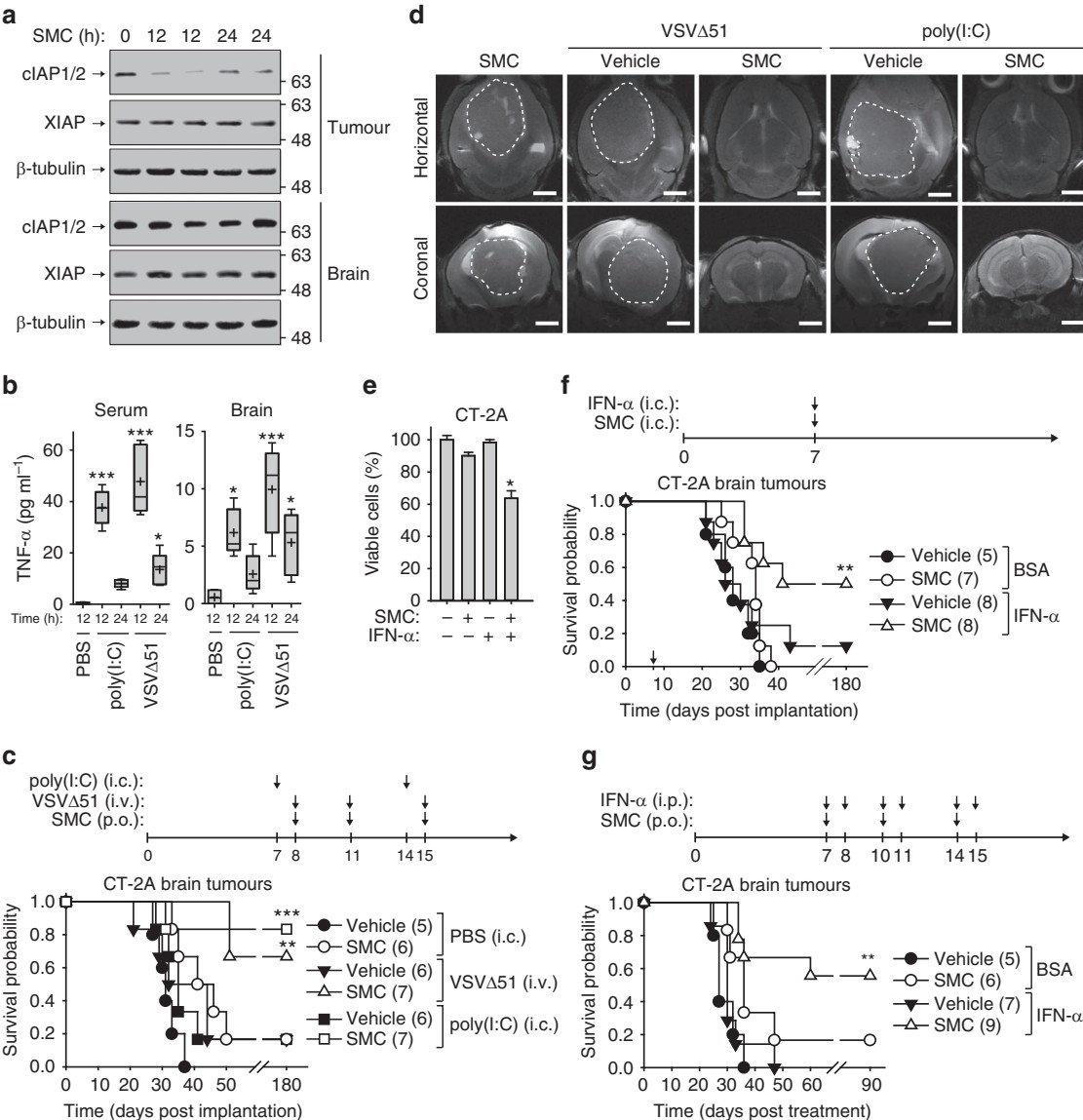

**Figure 2 | SMCs synergize with innate immunostimulants for the treatment of glioblastoma.** (**a**) Mice bearing intracranial CT-2A tumours were treated with 75 mg kg $^{-1}$ LCL161 (SMC, oral) and tumours and surrounding (non-tumorous) brain tissues were harvested at the indicated times for western blotting using the indicated antibodies. $n = 1$ for control and $n = 2$ for SMC-treated mice. (**b**) TNF-$\alpha$ levels from serum or solubilized brain homogenates was determined by ELISA of mice treated with PBS or 50 µg poly(I:C) i.p. or $5 \times 10^8$ PFU of VSV$\Delta$51 i.v. Crosses depict mean, solid horizontal lines depict median, boxes depict 25th to 75th percentile, and whiskers depict min–max range of the values. $n = 4$ for control mice and $n = 5$ for treated mice for each group. Significance was compared with PBS treatment using ANOVA with Dunnett's multiple comparison test. \*$P < 0.05$; \*\*\*$P < 0.001$. (**c**) Mice bearing intracranial tumours were treated at the indicated times with vehicle or 75 mg kg $^{-1}$ LCL161 (oral), PBS or 50 µg poly(I:C) i.c. or $5 \times 10^8$ PFU of VSV$\Delta$51 (i.v.). (**d**) Representative live MRI images from mice described in (**c**) were acquired at 40 days post implantation. Scale bar, 2 mm. (**e**) Alamar blue viability assay of CT-2A cells treated with vehicle or 5 µM LCL161 and 0.01% BSA or 1 µg ml $^{-1}$ IFN-$\alpha$B/D. Error bars, mean, s.d. $n = 4$. Representative data from three independent experiments using biological replicates. Statistical significance was compared with vehicle and BSA treatment using ANOVA using Dunnett's multiple comparison test. Significance is reported if $P < 0.0001$ (\*). (**f**) Mice bearing intracranial CT-2A tumours were treated i.c. at day 7 post-implantation with combinations of vehicle or 75 mg kg $^{-1}$ LCL161 and 0.01% BSA or 1 µg of IFN-$\alpha$B/D. (**g**) Mice bearing 7-days-old intracranial CT-2A tumours were treated with combinations of 75 mg kg $^{-1}$ LCL161 (oral) and BSA or 1 µg of IFN-$\alpha$B/D (i.p.). (**c,f,g**) Data represent the Kaplan–Meier curve depicting mouse survival. Log-rank with Holm–Sidak multiple comparison: \*\*$P < 0.01$; \*\*\*$P < 0.001$. Numbers in parentheses represent number of mice per group.

Interestingly, mice previously cured of orthotopic EMT6 mammary carcinomas by combined SMC treatments were completely resistant to tumour engraftment when rechallenged with EMT6 cells (Fig. 3b). However, another syngeneic cell line, 4T1, that shares the major histocompatibility proteins, was not rejected from these cured mice. We found that mice cured with intracranial CT-2A tumours were also resistant to tumour engraftment of CT-2A cells injected either subcutaneously or intracranially (Fig. 3c). We next evaluated the cytotoxic potential of CD8 T-cells from cured mice via an ELISpot assay. Stimulation of CD8 $^+$ T-cells from cured mice, but not cells isolated from naive mice, with CT-2A cells revealed the presence of specific reactive T-cells, as demonstrated by enhanced IFN-$\gamma$ and Granzyme B (GrzB) production (Fig. 4a). The inclusion of anti-PD-1 blocking antibodies further increased the expression of IFN-$\gamma$ and GrzB. Similar results were observed with mice cured

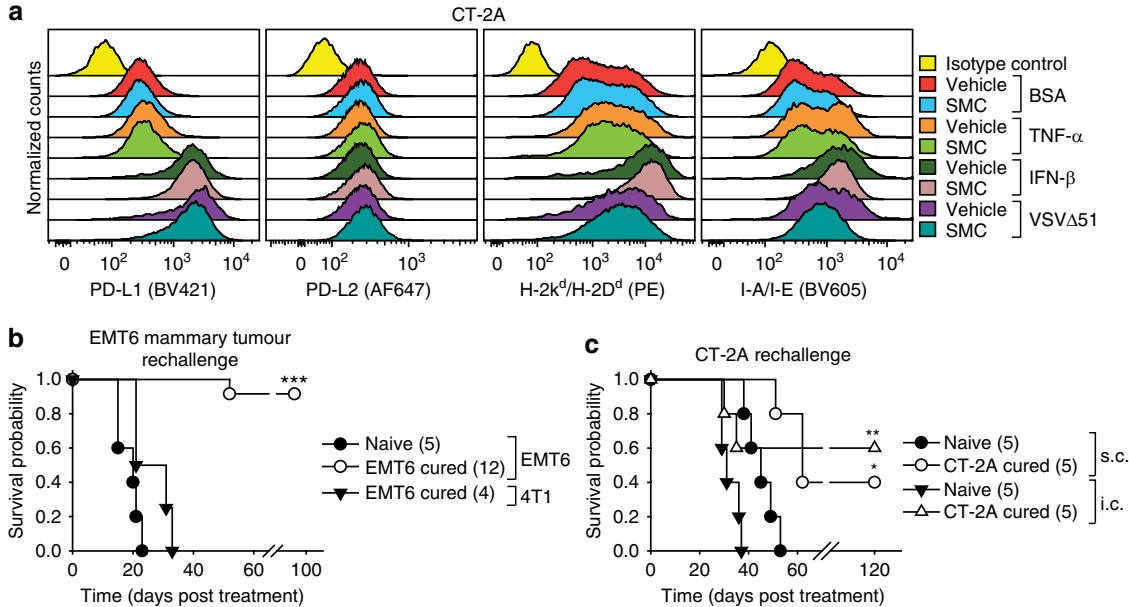

**Figure 3 | SMC-based combination treatment results in long-term immunological anti-tumour memory. (a)** CT-2A cells were treated for 24 h with vehicle or 5 µM LCL161 (SMC) and 0.01% BSA, 1 ng ml$^{-1}$ TNF-α, 250 U ml$^{-1}$ IFN-β or 0.1 MOI of VSVΔ51, and viable cells (Zombie Green negative) were analysed by flow cytometry using the indicated antibodies. Representative data from at three independent experiments using biological replicates. **(b,c)** Naïve mice or mice previously cured with SMC-based treatments of mammary fat pad EMT6 (mammary carcinoma, **b**) or intracranial CT-2A (glioblastoma, **c**) tumours were reinjected with EMT6 or mammary carcinoma 4T1 cells within the mammary fat pad or with CT-2A cells s.c. or i.c. Cells were implanted at 180 days initial post-implantation. Data represent the Kaplan–Meier curve depicting mouse survival. Log-rank with Holm–Sidak multiple comparison (compared with method of implantation): *$P<0.05$; **$P<0.01$; ***$P<0.001$. Numbers in parentheses represent number of mice per group.

of EMT6 tumours (Supplementary Fig. 9). Collectively, these results suggest the generation of a robust and specific long-term tumour immunity using SMC combination therapy.

**Immune checkpoint inhibitors synergize with IAP antagonists.** We next investigated whether a current class of cancer immunotherapy, known as immune checkpoint inhibitors or ICIs, could enhance SMC efficacy. It has been recently reported that ICI treatment of glioblastoma in mice results in at least a partial extension of survival[26,27]. We first sought to determine whether SMC treatment influences PD-1 expression in a subset of infiltrating immune cells within CT-2A brain tumours. While there was no statistical difference between the levels of infiltrating CD3$^+$ or CD3$^+$ CD8$^+$ cells within intracranial CT-2A tumours, we observed a robust increase of CD3$^+$ and CD3$^+$ CD8$^+$ cells expressing the immune checkpoint, PD-1 (Fig. 4b; Supplementary Fig. 10). Although there was a general increase in the expression of PD-L1 in CD45$^-$ cells, which are predominantly CT-2A cells, the trend did not reach statistical significance (Fig. 4c).

To determine whether the increased levels of PD-1$^+$ CD8$^+$ T-cells may be a negative modulator for SMC efficacy, we also assessed blocking the checkpoint target, PD-1, as well as CTLA-4, in combination with SMC using two mouse models of glioblastoma. The systemic administration of anti-PD-1 or anti-CTLA4 antibodies demonstrated no activity on their own (Fig. 4d,e). In contrast, the combination of anti-PD-1 and SMC significantly extended survival and resulted in 71% and 33% durable cure rates in the CT-2A and GL261 models, respectively. Furthermore, when combined with a SMC, the anti-PD-1 biologic was superior to the anti-CTLA4 biologic in the CT-2A model (71% versus 43%) (Fig. 4d). There are two structural classes of SMCs: monomers and dimers. Monomeric SMCs consist of a single chemical molecule that binds to the BIR domains of the IAPs while dimeric SMCs consist of two SMC molecules

connected by a linker allowing for cooperative binding and/or tethering of IAPs. A clinically advanced SMC, LCL161, is the focus of most of our studies, and is a potent monomer. We next sought to assess whether another clinically advanced dimeric SMC similarly synergizes with an ICI for the treatment of glioblastoma. Similar to our previous results, we observed a significant increase in survival of mice bearing intracranial CT-2A tumours when treated with anti-PD-1 and the dimer SMC, Birinapant (Fig. 4f). As the combined blockade of PD-1 or CTLA-4 is beneficial for patients with melanoma[28], we sought to determine whether the combination of anti-PD-1 and anti-CTLA-4 would similarly significantly enhance SMC therapy. Consistent with a previous report, the combination of antibodies targeting PD-1 and CTLA-4 was effective at inducing durable cures in a mouse model of cancer[27], as we observed an overall survival rate of 67% (Fig. 4g). Strikingly, the inclusion of SMC treatment with anti-PD-1 and anti-CTLA-4 together resulted in a 100% durable cure rate.

The synergistic effect between SMC and ICIs is not restricted to brain tumours. We also observed a significant extension of the survival of mice bearing a highly aggressive and treatment refractory model of multiple myeloma using MPC-11 cells (Supplementary Fig. 11). A durable cure rate of 75% was also obtained in mice harbouring mammary EMT6 tumours, which was further increased to 100% with the inclusion of an immune stimulant (Supplementary Fig. 12).

**CD8$^+$ T-cells are required for efficacy of SMCs and ICIs.** To provide an initial insight into the cellular mechanism of action, we profiled the production of immune factors from CT-2A cells that were cocultured with splenocytes derived from mice cured of intracranial CT-2A tumours using combined SMC and anti-PD-1 treatment. Similar to that depicted in Fig. 4a and Supplementary Fig. 9, we observed a significant increase in the production of IFN-γ and GrzB from CT-2A cells co-incubated with splenocytes

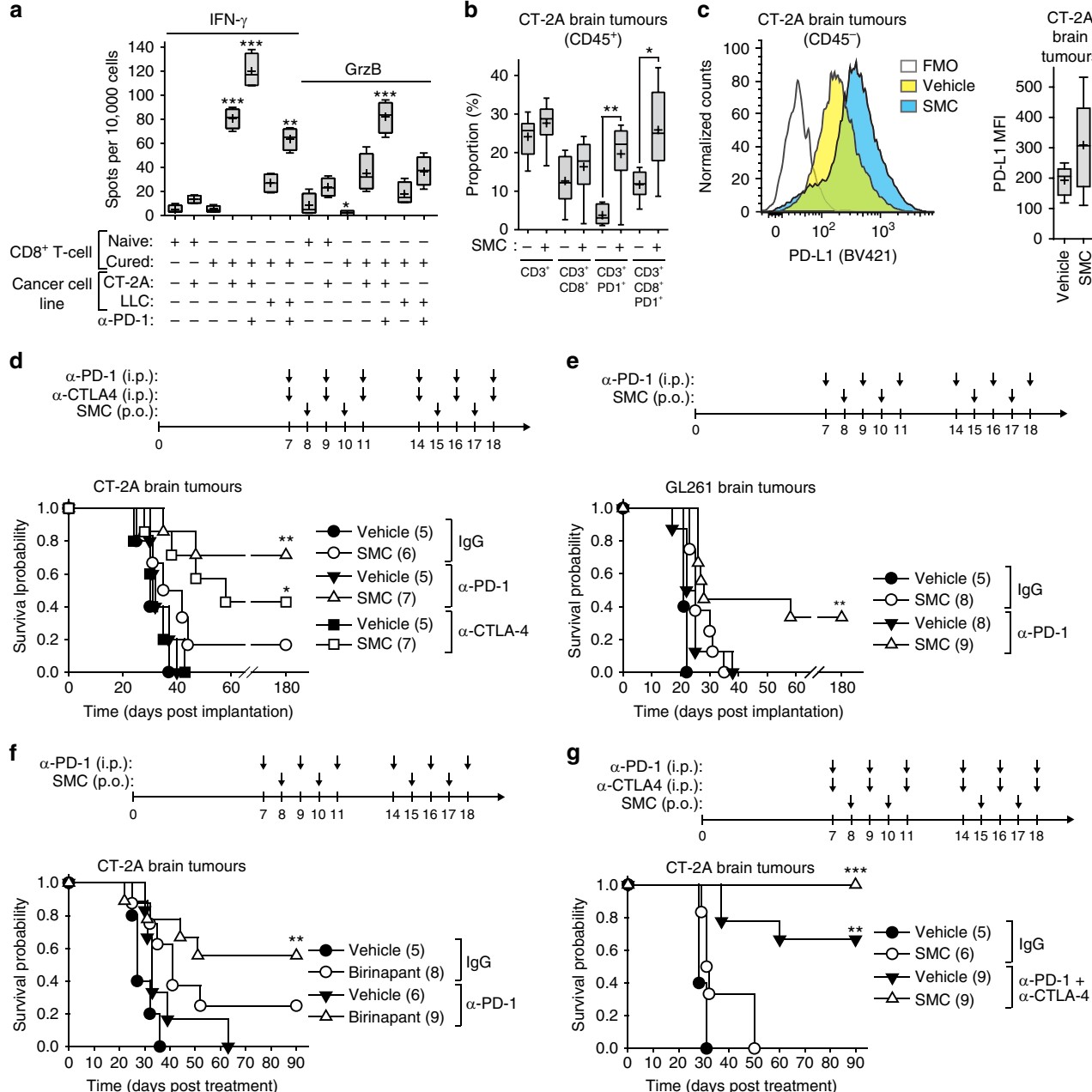

**Figure 4 | SMCs synergize with antibodies targeting immune checkpoints in mouse models of glioblastoma.** (**a**) Splenic CD8[+] T-cells were enriched from naive mice or mice previously cured of CT-2A tumours, and subjected to ELISpot assays for the detection of IFN-γ and GrzB. Cancer cells (CT-2A, LLC) were cocultured with CD8[+] cells (25:1 ratio) and 10 µg ml[−1] of control IgG or α-PD-1 for 48 h. $n = 4$ of mice per group. Significance was compared with naive CD8[+] T-cell co-incubated with CT-2A cells as assessed by ANOVA with Dunnett's multiple comparison test. $*P < 0.05$; $**P < 0.01$; $***P < 0.001$. (**b**) Mice bearing intracranial CT-2A tumours were treated with 75 mg kg[−1] LCL161 orally (SMC) on post-implantation day 14, 16, 21 and 23. Viable cells from tumour masses were analysed by flow cytometry for the detection of CD45 (BV605), CD3 (APC-Cy7), CD8 (PE) and PD-1 (BV421). Representative plots are shown in Supplementary Fig. 6. Statistical significance for each pair was assessed by a $t$-test. $*P < 0.05$; $**P < 0.01$. (**c**) Viable tumour cells from the experiment in **b** were analysed by flow cytometry using the antibodies CD45 (PE) and PD-L1 (BV421). $n = 6$ of mice per group. FMO, fluorescence minus one. Statistical significance was assessed by a $t$-test. (**d**–**g**) Mice bearing intracranial CT-2A (**d,f,g**) or GL261 (**e**) tumours were treated at the indicated times with combinations of vehicle, 75 mg kg[−1] LCL161 orally (**d,e,g**) or vehicle or 30 mg kg[−1] Birinapant i.p. (**f**) and 250 µg of IgG, α-PD-1 or α-CTLA-4 (i.p.) or both combined (**g**). Data represent the Kaplan–Meier curve depicting mouse survival. Log-rank with Holm–Sidak multiple comparison: $*P < 0.05$; $**P < 0.01$; $***P < 0.001$. Numbers in parentheses represent number of mice per group. (**a**–**c**) Crosses depict mean, solid horizontal lines depict median, boxes depict 25th to 75th percentile, and whiskers depict min–max range of the values. (**d**) Representative data from two independent experiments.

derived from surviving mice (Fig. 5a; Supplementary Fig. 13a). Notably, there was an increase in the production of Interleukin 17 (IL-17), which has been shown to promote an antitumor cytotoxic T-cell response[29]. We also observed a reduction in the expression of the proinflammatory cytokines IL-6 and TNF-α,

which was unexpected, given that IL-17 has been previously found to stimulate the NF-κB pathway[30]. However, the expression of IFN-γ and IL-17 from splenocytes isolated from cured mice significantly increased with anti-PD-1 or PD-L1 treatment, suggesting that a T-cell-based immune response can be

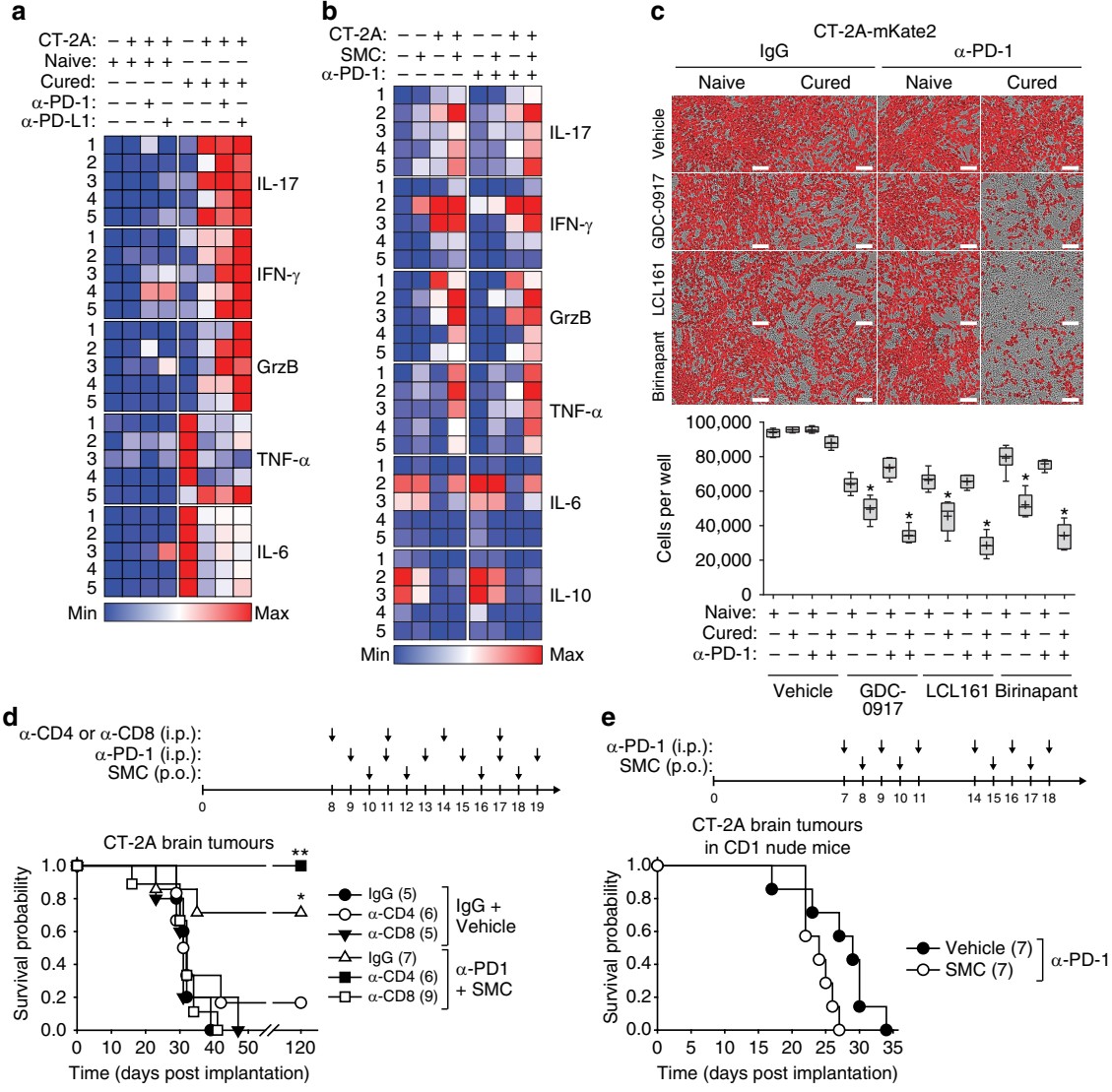

**Figure 5 | CD8⁺ T-cells are required for synergy between SMC and immune checkpoint inhibitors for the treatment of glioblastoma. (a)** The expression of the indicated immune factors was detected by ELISA from cell culture supernatants of CT-2A cells that were co-incubated for 48 h with splenocytes derived from naive mice or mice previously cured of intracranial CT-2A tumours by SMC and anti-PD-1 cotreatment (1:20 ratio of CT-2A cells to splenocytes). Data are plotted as heat maps using normalized scaling. Box and whisker plots of the data are shown in Supplementary Fig. 13a. **(b)** Quantification of the indicated factor was determined by ELISA from CT-2A cells that were cocultured with splenocytes derived from naive or cured mice (1:20 ratio) and treated with vehicle or 5 μM LCL161 (SMC) for 48 h. Box and whisker plots of the data are shown in Supplementary Fig. 13b. **(c)** Splenocytes from naive or cured mice were cocultured with mKate2-tagged CT-2A cells (CT-2A-mKate2) in the presence of 20 μg ml⁻¹ control IgG or anti-PD-1 and 5 μM of the indicated SMC. Enumeration of CT-2A-mKate2 cells was performed using the Incucyte Zoom. Crosses depict mean, solid horizontal lines depict median, boxes depict 25th to 75th percentile, and whiskers depict min–max range of the values. Significance was compared with naive splenocytes as assessed by ANOVA with Dunnett's multiple comparison test. Significance is reported as * when $P < 0.0001$. $n = 6$ for naive mice and $n = 6$ for cured mice. Scale bar, 100 μm. **(d)** C57BL/6 mice harbouring intracranial CT-2A tumours were treated at the indicated date with combinations of either IgG (i.p.) and vehicle (oral) or α-PD-1 (i.p) and 75 mg kg⁻¹ LCL161 (oral) and i.p. administration of either IgG, α-CD4 or α-CD8 (all antibodies were 250 μg). **(e)** CD-1 nude mice bearing intracranial CT-2A tumours were treated at the indicated times with combinations of vehicle or 75 mg kg⁻¹ LCL161 orally and PBS or 250 μg of IgG or α-PD-1 i.p. **(d,e)** Data represent the Kaplan–Meier curve depicting mouse survival. Log-rank with Holm–Sidak multiple comparison: *$P < 0.05$; **$P < 0.01$. Numbers in parentheses represent number of mice per group.

augmented on checkpoint inhibition through the PD-1 axis. We next sought to determine whether this cytokine response is affected by SMC treatment. Among the previously analysed cytokines, the inclusion of SMC in these cocultures along with anti-PD-1 blockade increased the secretion of IFN-γ, GrzB, IL-17 and TNF-α (Fig. 5b; Supplementary Fig. 13b). Notably, the level of IL-6 in the supernatant was not affected by SMC treatment. Furthermore, the immunosuppressive cytokine IL-10 had a general trend of decreased secretion with combined SMC and anti-PD-1 treatment. As there is an increase in the levels of GrzB, a cytotoxic factor that is partially blocked by XIAP[31–33], and TNF-α, we next assessed whether cocultures of glioblastoma cells with splenocytes from naive mice or mice previously cured of CT-2A intracranial tumours would lead to death of CT-2A cells. Using differently structured SMCs, we saw a statistically significant increase in the death of CT-2A cells in the presence of SMCs, and this response was increased with the inclusion of anti-PD-1 antibodies (Fig. 5c).

Collectively, these results indicate that a robust effector T-cell response is elicited with the combination treatment of ICI and SMC. To further elucidate the cellular mechanism of action, we undertook the depletion of immune cells using specific CD4 or CD8 targeting antibodies. We found that the 71% cure rate induced by the combination therapy is completely abrogated on depletion of CD8 $^+$ T-cells (Fig. 5d). Interestingly, the depletion of CD4 $^+$ T-cells resulted in a 100% cure rate with the combination of SMC and anti-PD-1, and a 17% cure rate in the control group. These results suggest that removal of CD4 $^+$ immunosuppressive cells (such as regulatory T-cells) aids with the induction of tumour regression and that CD4 $^+$ cells are not required for efficacy of the combined treatment approach[34]. In a second approach, intracranial CT-2A tumours were established in CD-1 nude mice, which lack functional T-cells, and then treated with the combination of anti-PD-1 antibodies with vehicle or SMC. The survival advantage provided by the SMC and anti-PD-1 combination was lost in these T-cell deficient mice (Fig. 5e). Overall, the synergistic effect between SMC and anti-PD-1 is dependent on a functional adaptive immune response and thus implicates CD8 $^+$ T-cells as the primary immune cell mediators for long-term *in vivo* efficacy.

**SMC treatment affects intratumoral immune cell infiltration**. To understand the immune cellular aspect of the synergy between SMC and ICI treatment, we evaluated the profiles of infiltrating CD45 $^+$ immune cells in mice bearing glioblastoma. In these studies, we evaluated the infiltrating immune cells in later stage glioblastoma tumours following anti-PD-1 and SMC cotreatment (Fig. 6a). A flow cytometry analysis of tumour infiltrating T-cells revealed a statistically insignificant trend in the proportion of CD4 $^+$ and CD8 $^+$ T-cells between the vehicle and IgG control treatment group and all single and double-treated mice (Fig. 6b). However, an analysis of CD4 $^+$ and CD25 $^+$ T-cells, indicative of a regulatory T-cell (Treg) population[35], revealed a significant decrease of this cell population with SMC treatment alone or combination of SMC and ICI (Fig. 6c). Next, we characterized the surface presentation of PD-1 in T-cells following single and combinatorial treatment. As previously observed (Fig. 4b), we noted a significant increase in CD8 $^+$ T-cells expressing PD-1 in mice treated with SMC alone, and the treatment of anti-PD-1 or combined treatment of SMC and anti-PD-1 resulted in less detectable surface presentation of PD-1 (Fig. 6d). In addition, we observed a trend in the decreased presentation of PD-1 in CD4 $^+$ T-cells in SMC or anti-PD-1 treatment groups. However, the detectable level of surface PD-1 was abrogated with combinatorial treatment of SMC and anti-PD-1 (Fig. 6e).

In addition to the observed T-cell infiltration of intracranial glioblastoma tumours, we next characterized the presence of myeloid-derived suppressor cells (MDSC) and astrocytes/microglia. In contrast to a previous report[27], we did not detect differences in the MDSC population (CD11b $^+$ Gr1 $^+$) in any treatment cohorts (Fig. 6f). However we noted that the astrocyte/microglia population was significantly decreased in the treatment cohorts that included anti-PD-1 (Fig. 6g). Overall, these results indicate that the consequence of combinatorial treatment is the decrease of an immunosuppressive CD4 $^+$ T-cell population with a concomitant decrease of PD-1 presentation in T-cells and a reduction of astrocytes and/or microglia.

**SMC synergy with ICIs is dependent on TNF-α**. We next characterized the tumoral cellular cytokine and chemokine profiles of mice bearing intracranial glioblastoma tumours treated with combinations of SMC and anti-PD-1. Flow cytometry

analysis revealed that there was an increase of CD8 $^+$ cells expressing GrzB with the inclusion of anti-PD-1 antibodies. The ratio of cytotoxic CD8 $^+$ (Fig. 7a) and CD4 $^+$ Treg ratio was also increased in the anti-PD-1 and SMC and anti-PD-1 treatment cohorts (Fig. 7b). In addition to assessing GrzB expression, we analysed the levels of IFN-γ and TNF-α in T-cells. Unexpectedly, we observed a decrease in the proportion of CD4 $^+$ cells expressing IFN-γ on SMC treatment (even in inclusion of antibodies targeting PD-1) but saw no change in the expression level of IFN-γ in any treatment cohort within CD8 $^+$ cells (Fig. 7c). As a main mode of cell death mediated by SMCs relies on the engagement of the apoptotic pathways in cancer cells on stimulation by TNF-α, we accordingly analysed the expression level of TNF-α in T-cells. In this context, we observed a significant increase of TNF-α expressing CD4 $^+$ and CD8 $^+$ T-cells (Fig. 7d), indicating that these T-cells can directly induce SMC-mediated tumour cell death.

We also evaluated the effect of combined SMC treatment and anti-PD-1 blockade on serum concentration and gene expression levels of cytokines and chemokines in the intracranial CT-2A glioblastoma model. We detected statistically significant increases in the proinflammatory cytokines IFN-β, IL-1-α, IL-1β, IL-17 and the multifaceted cytokines IFN-γ, IL-27 and GM-CSF (Fig. 7e; Supplementary Fig. 14). Notably, there was no difference in the presence of anti-inflammatory cytokines such as IL-10. Similarly, an analysis of the cytokine and chemokine expression profiles within intracranial CT-2A tumours following combined SMC and ICI treatment revealed clustering of proinflammatory cytokines and chemokines (Fig. 7f; Supplementary Fig. 15). Among these candidates from SMC or combined SMC and ICI treatment were the proinflammatory cytokines *IFN-β*, *IL-1β*, *IL-17*, *Osm* and *TNF-α*, the chemokines *Ccl2* (also known as *MCP-1*), *Ccl5*, *Ccl7*, *Ccl22*, *Cxcl9*, *Cscl10* and *Cxcl11*, and multifaceted factors such as *FasL*, *IL-2*, *IL-12* and *IFN-γ*. As we observed a consistent increase in the levels of IFN-β and IFN-γ, we next sought to characterize the functional role of these signalling molecules with the use of blocking/neutralizing antibodies in mice bearing intracranial CT-2A tumours and treated with SMC and anti-PD-1. Abrogation of type I IFN signalling by using an antibody that blocks the IFNAR1 receptor negated the synergistic effects towards increasing survival of mice bearing intracranial CT-2A tumours following combined SMC and anti-PD-1 treatment (Fig. 7g). In contrast, antagonism of IFN-γ function by employing an anti-IFN-γ antibody partially inhibited the synergistic effects of combined SMC and ICI treatment. Overall, these results indicate that each treatment agent, including when combined, results in the generation of different gene and protein signatures, but overall, is dependent on intact type I IFN signalling.

Overall, our results indicate that the synergistic effects between SMC and ICI can be primarily attributed towards enhancing a CTL-mediated attack against glioblastoma cells, and this involves a proinflammatory response that includes type I IFN. The coculture of CT-2A cells and CD8 $^+$ T-cells isolated from mice previously cured of intracranial tumours resulted, as expected, in an increase of GrzB positive CD8 $^+$ T-cells, which was not increased with SMC treatment alone (Fig. 8a). However, there was only a slight decrease of viable CT-2A cells when co-incubated with the same CTLs, even when the PD-1/PD-L1 axis was abrogated (Fig. 8b). As we previously noted that the type I IFN response also leads to the production of TNF-α[16], we assessed the ability of T-cells to produce TNF-α following SMC treatment in the presence of glioblastoma cells. Accordingly, we next evaluated the production of TNF-α. The inclusion of SMC significantly increased the proportion of CD8 $^+$ T-cells expressing TNF-α, regardless of inclusion of antibodies targeting PD-1 (Fig. 8a). In accordance with the increased expression level of

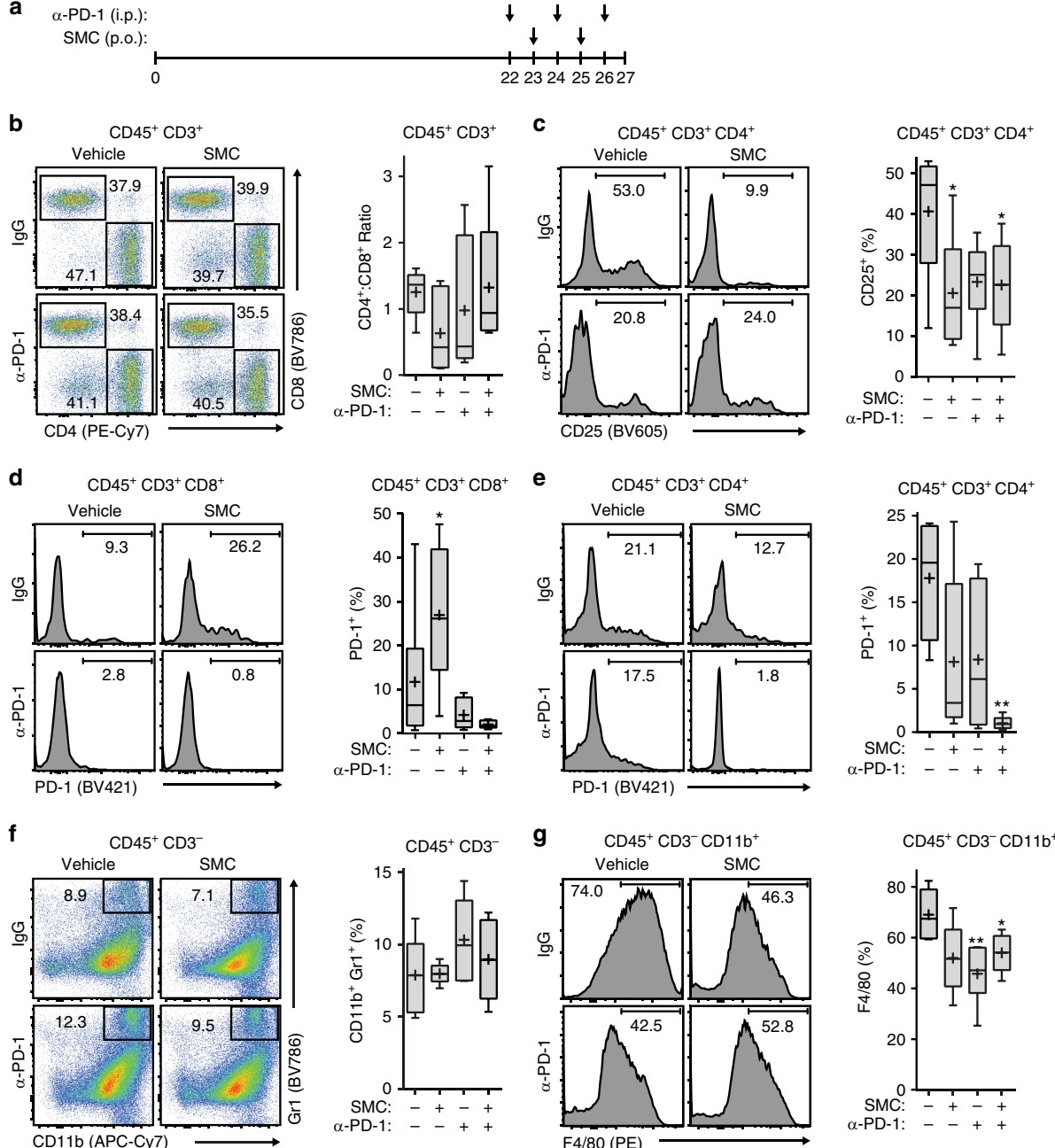

**Figure 6 | SMC and immune checkpoint inhibitor treatment in mouse models of glioblastoma leads to changes in immune effector cell infiltration.**
(**a**) Mice bearing intracranial CT-2A tumours were treated at the indicated times with vehicle or 75 mg kg$^{-1}$ LCL161 orally (SMC) and 250 µg IgG or anti-PD-1 i.p. Mice were euthanized on day 27 post implantation. (**b–e**) Viable T-cells isolated from tumours were processed for flow cytometry using the following antibodies: CD45 (PE-Cy5), CD3 (APC), CD4 (PE-Cy7), CD8 (BV786), CD25 (BV605) and PD-1 (BV421). (**f,g**) Viable cells from the experiment in (**a**) were processed for flow cytometry using the following antibodies: CD45 (BV605), CD11b (APC-Cy7), Gr1 (BV786), F4/80 (PE) and CD3 (APC). All panels: Crosses depict mean, solid horizontal lines depict median, boxes depict 25th to 75th percentile, and whiskers depict min–max range of the values. Significance was compared with vehicle and IgG-treated mice as assessed by ANOVA with Dunnett's multiple comparison test. *$P < 0.05$; **$P < 0.01$. $n = 6$ for each treatment group.

TNF-α from CD8$^+$ T-cells, we observed significant decrease of CT-2A cells in a coculture system using CT-2A cells and CD8$^+$ T-cells from cured mice (Fig. 8b). Notably, the SMC-mediated effects on eliciting death of CT-2A cells were mainly dependent on TNF-α (the primary mediator of SMC-induced tumour killing). Next, we evaluated whether SMC treatment enhances T-cell proliferation. Indeed, we observed a significant decrease of CFSE-loaded CD8$^+$ T-cells, along with the appearance of a new population of faintly labelled CFSE-cells, following co-incubation

of CT-2A cells, and this effect was pronounced with the inclusion of SMC and anti-PD-1 (Supplementary Fig. 16).

These results indicate that cytotoxic T-cells, in response to SMC and anti-PD-1 treatment, may lead to enhanced tumour cell death due to the increased production of GrzB and TNF-α, pro-death factors that induce tumour cell death due to the antagonism of the IAPs. We functionally characterized the role of TNF-α by employing blocking antibodies targeting TNF-α. When systemic blockade of TNF-α was applied, we observed almost a complete

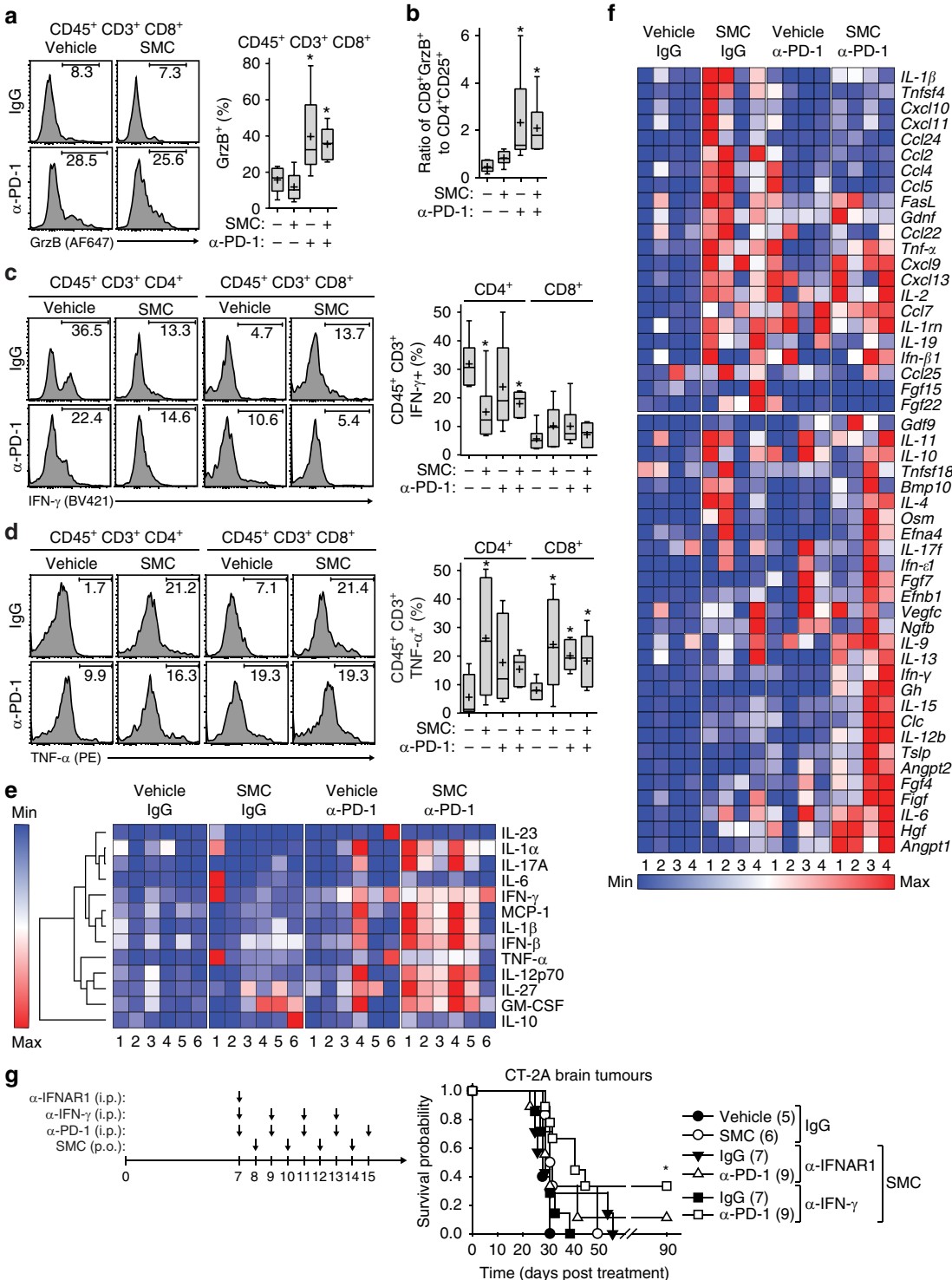

**Figure 7 | SMC and immune checkpoint inhibitor combination induces a proinflammatory cytokine response and efficacy is dependent on type I IFN.**
(**a–d**) Mice were treated as in Fig. 6a, and viable cells from brain tumours were isolated and processed for flow cytometry using the following antibodies: CD45 (BV605), CD3 (APC-Cy7), CD4 (PE-Cy7), CD8 (BV786/0), IFN-γ (BV421), TNF-α (PE) and GrzB (AF647). Crosses depict mean, solid horizontal lines depict median, boxes depict 25th to 75th percentile, and whiskers depict min–max range of the values. Significance was compared with vehicle and IgG-treated mice as assessed by ANOVA with Dunnett's multiple comparison test. *$P < 0.05$. $n = 6$ for each treatment group. (**e**) Serum from mice depicted in Fig. 6a was processed for multiplex ELISA for the quantitation of the indicated proteins. Data are plotted as heat maps using normalized scaling. Representation of the same data as box and whisker plots is in Supplementary Fig. 14. $n = 6$ for each treatment group. (**f**) Mice were treated as in Fig. 6a and intracranial CT-2A tumors were processed for quantitation of 176 cytokine and chemokine genes by RT-qPCR. Shown are normalized heat maps of two major groups identified by hierarchical clustering. The complete hierarchical cluster profiles are in Supplementary Fig. 15. $n = 4$ for each treatment group. (**g**) Mice bearing intracranial CT-2A tumours were treated at the indicated post-implantation day with vehicle or 75 mg kg$^{-1}$ LCL161 (oral) or i.p. with the relevant isotype IgG control or 2.5 mg α-IFNAR1, 350 μg α-IFN-γ or 250 μg α-PD-1. Data represent the Kaplan-Meier curve depicting mouse survival. Log-rank with Holm-Sidak multiple comparison: *$P < 0.05$. Numbers in brackets denote the size of the treatment groups.

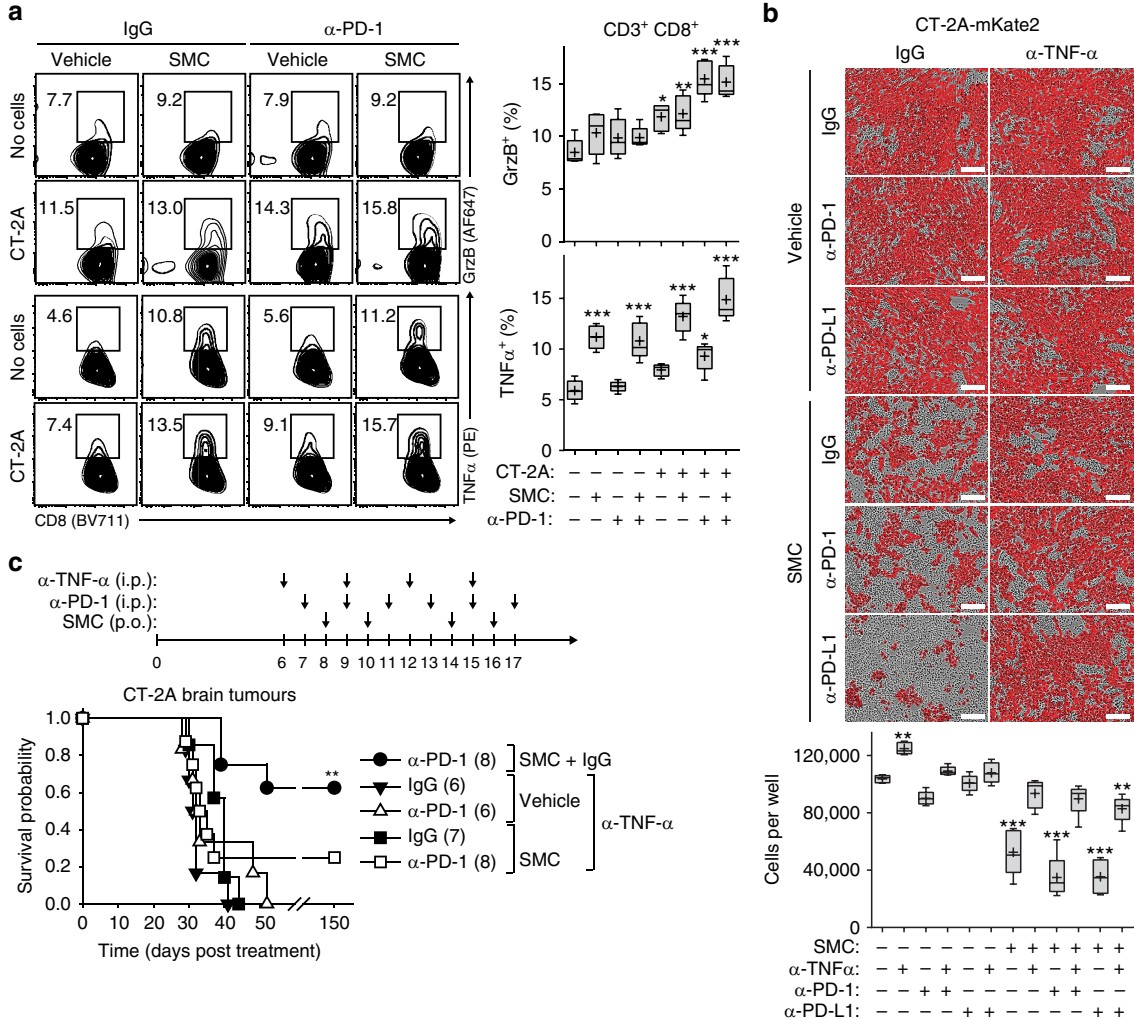

**Figure 8 | The proinflammatory cytokine TNF-α is required for T-cell-mediated death upon SMC and immune checkpoint inhibitor treatment.**
(**a**) Isolated CD8 T-cells derived from the spleen and lymph nodes from mice previously cured of intracranial CT-2A tumours were cocultured with CT-2A cells in the presence of vehicle or 5 μM LCL161 and 20 μg ml$^{-1}$ isotype-matched IgG or α-PD-1 for 24 h. Viable T-cells were processed for flow cytometry using the following antibodies: CD3 (APC-Cy7), CD8 (BV711), GrzB (AF647) and TNF-α (PE). Crosses depicts mean, solid horizontal line depicts median, box depicts 25th to 75th percentile, and whiskers depicts min–max range of the values. Significance was compared with vehicle and IgG-treated mice as assessed by ANOVA with Dunnett's multiple comparison test. *$P < 0.05$; **$P < 0.01$; ***$P < 0.001$. $n = 5$ for each treatment group. (**b**) CD8$^+$ T-cells were cocultured with mKate2-tagged CT-2A cells (CT-2A-mKate2) for 72 h in the presence of vehicle or 5 μM LCL161 and 20 μg ml$^{-1}$ of control IgG, α-PD-1 or α-TNF-α. Enumeration of mKate2-positive cells was acquired using the Incucyte Zoom software. Crosses depict mean, solid horizontal lines depict median, boxes depict 25th to 75th percentile, and whiskers depict min–max range of the values. Significance was compared with vehicle and IgG-treated mice as assessed by ANOVA with Dunnett's multiple comparison test. **$P < 0.01$; ***$P < 0.001$. $n = 5$ for each treatment group. Scale bar, 100 μm. (**c**) Mice bearing intracranial CT-2A tumours were treated at the indicated post-implantation day with vehicle or 75 mg kg$^{-1}$ LCL161 (oral) or i.p. with the relevant isotype IgG control or 500 μg α-TNF-α or 250 μg α-PD-1. Data represent the Kaplan-Meier curve depicting mouse survival. Log-rank with Holm-Sidak multiple comparison: **$P < 0.01$. Numbers in parentheses represent number of mice per group.

reversal of the efficacy of combined SMC and ICI treatment (Fig. 8c), highlighting the importance of TNF-α for the synergistic effect of these disparate agents.

## Discussion

SMCs are a class of molecularly targeted drugs that are being evaluated as an anti-cancer therapy. The immunomodulatory anti-cancer effects of SMCs are multimodal (Fig. 9; Supplementary Fig. 17). SMCs can polarize macrophages away from the immunosuppressive M2 type towards the inflammatory TNF-α-producing M1 phenotype[36]. Moreover, SMC anti-cancer effects are highly potentiated by proinflammatory cytokines, and

the presence of these cytokines, such as TNF-α or TRAIL, within the tumour microenvironment leads to tumour cell death[16,37–39]. Specifically, SMC-mediated depletion of the cIAPs converts the TNF-α-mediated survival response into a death pathway in cancer cells. Recently, we demonstrated that a self-limiting cytokine storm that includes TNF-α and TRAIL can be generated with immunostimulatory agents to safely synergize with SMCs to cure mice of cancer and that this outcome depends largely on innate immunity[16]. Similarly, the delivery of cell-permeable Smac peptides with recombinant TRAIL was found to lead to a significant increase in survival of mice bearing intracranial glioblastoma tumours[40]. In addition, the cIAPs also regulate the activation of the alternative NF-κB pathway in immune cells,

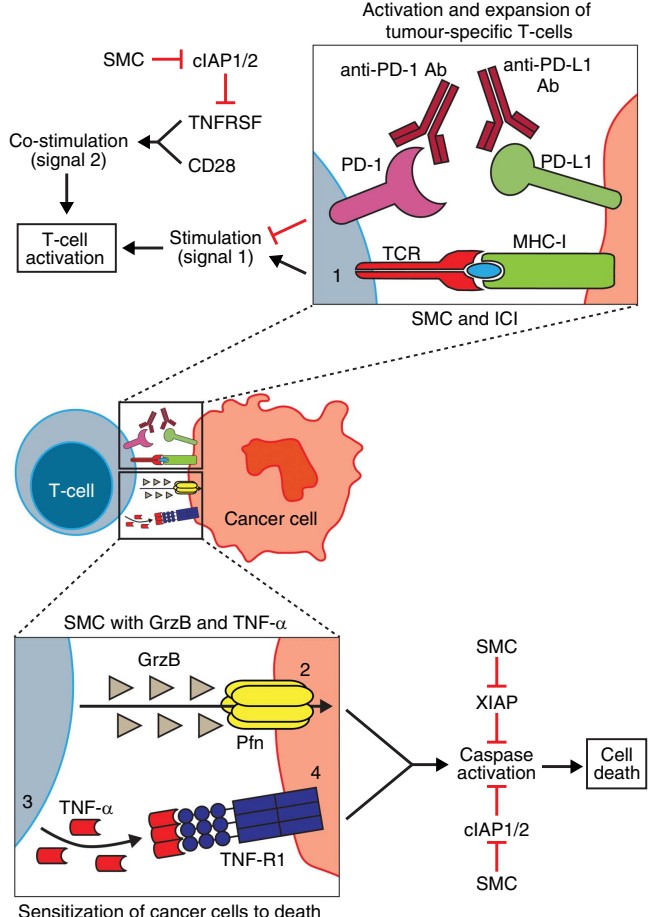

**Figure 9 | Cooperative and complimentary mechanisms for synergy between SMCs and ICI.** (1) The presence of therapeutic recombinant antibodies that block the PD-1/PD-L1 axis allows for signalling of the TCR of a CD8$^+$ T-cell with its associated antigen presented by the cancer cell through a MHC-I molecule. Concurrent depletion of the IAPs through SMC treatment can enhance T-cell activation, likely by providing a tumour necrosis factor receptor superfamily (TNFRSF) co-stimulatory response (similar to 4-1BB or OX40 activation), resulting in enhanced activation and expansion of tumour-specific CD8$^+$ T-cells. As a result, Granzyme B (GrzB) and Perforin (Pfn) are secreted to kill target cells. (2) SMC-mediated antagonism of the casp-3 inhibitor, XIAP, can result in enhanced death of tumour cells by GrzB. (3) The depletion of cIAP1 and cIAP2 by SMCs leads to increased local production of TNF-α by T-cells in the tumour microenvironment, an effect that is likely mediated by activation of the alternative NF-κB pathway. (4) As a result of cIAP1/ 2 loss, SMC-treated cancer cells are sensitized to cell death induction in the presence of proinflammatory cytokines, such as TNF-α.

which is postulated to produce co-stimulatory signals to enhance adaptive immune responses against tumours[5,10,41]. In support of this concept, it is relevant that SMCs stimulate dendritic cell maturation[42] and T-cell proliferation, and potentiate cytotoxic T-cell activation and expansion in the absence of ligands such as 4-1BB[5,43]. Notably, these SMC-mediated effects have also been shown to increase T-cell activity of a cancer vaccine approach[5]. Our current studies demonstrate that SMCs can cooperate and dramatically intensify the action of ICIs, including anti-PD-1 or anti-CTLA-4 antibodies, allowing for durable cures of mice bearing aggressive intracranial tumours. The multiplicity and complexity of mechanisms involved with SMC therapy make it difficult to isolate the individual roles for the varied immunomodulatory actions in the combination synergy.

However, it is clear from this and previous work that TNF-α cytotoxicity is involved. Moreover, the current study further demonstrates that CD8$^+$ T-cells are also required for anti-cancer activity when an ICI is combined with an SMC.

We chose glioblastoma as a model to study because it represents a high-fatality cancer for which immunotherapy is being investigated as a potential therapy. In addition, cIAP1 and cIAP2, which are co-amplified in some brain tumours[18], represent valid drug targets for glioblastoma. Brain cancer also raises additional challenges for testing cancer therapies, as drug-penetration across the BBB may be an impediment to drug activity and differences in the brain immune system may affect tumour immunotherapies. We show that the oral administration of the monomer SMC, LCL161 (ref. 44) can induce the antagonism of its IAP targets within brain tumours, yet does not lead to downregulation of the IAP proteins in normal (non-tumorous) brain tissues. The penetrance of SMCs may be explained by the compromised BBB which is seen in brain tumours. In addition, we find that systemically applied immunostimulatory agents such as oncolytic VSV or recombinant IFN-α can result in increased TNF-α in the brain to potentially kill tumours with SMCs, releasing tumour antigens and stimulating inflammatory and immunomodulatory responses. Our results are in agreement with a report demonstrating that systemic infection or immune stimulation induces cytokine production including type I IFN within the brain[23]. Importantly, the systemic administration of antibody-based drugs targeting ICIs have been found to induce tumour responses for intracranial glioblastoma in mouse models and for humans with melanoma brain metastases[26,27,45,46]. It may be that these antibodies may infiltrate into brain tumours or that the antibodies coat T-cells in the periphery blocking the inhibitory action of immune checkpoint molecules once they extravagate from the circulation into brain tissue and then into the tumour microenvironment[47].

We and others shown that SMCs or Smac peptides kill cancer cells when combined with immunostimulatory agents that induce IL-1β, IFN-β, TNF-α and TRAIL[16,31–33,40]. In this report, we show for the first time that SMCs can potentiate the activity of ICIs in mouse tumour models. Furthermore, this combination effect depends on the presence of CD8$^+$ T-cells with a concomitant decrease of immunosuppressive CD4$^+$ T-cells, and on the type I and II IFN and TNF-α signalling pathways, clearly implicating the role of adaptive immunity for SMC-mediated cures in mice. We propose that SMC-mediated T-cell co-stimulatory signals provide the drive for adaptive immune responses that develop against the tumour and this is fully realized when the brakes imposed by co-inhibitory signals such as PD-1 or PD-L1 are removed with ICIs. Although, this mechanistic hypothesis remains to be firmly established, the fact that SMCs can kill cancer cells by at least two different mechanisms (involving both the innate and the adaptive immune system) makes these immunomodulatory drugs a compelling and rational class of agents that should be tested in a combination immunotherapy clinical setting. Indeed, cancer trials testing SMC combination immunotherapies, in particular with immune checkpoint inhibitors, are expected to initiate soon (ClinicalTrials.gov: NCT02587962, NCT02890069). We eagerly await the initiation of these trials as we continue to decipher the many SMC-mediated immunomodulatory effects that potentiate the killing of tumour cells.

## Methods

**Reagents.** Novartis provided LCL161 (ref. 44). Tetralogic Pharmaceuticals provided Birinapant[48]. AT-406, GDC-0917 and AZD-5582 were purchased from Active Biochem. TNF-α was purchased from Enzo. IFN-β was obtained from PBL

Assay Science. Broad host range IFN-αB/D was produced in yeast and purified by affinity immunochromatography[24]. Nontargeting siRNA or siRNA targeting cFLIP were obtained from Dharmacon (ON-TARGETplus SMARTpool). High molecular weight poly(I:C) was obtained from Invivogen.

**Cell culture.** Cells were maintained at 37 °C and 5% $CO_2$ in DMEM media supplemented with 10% heat-inactivated fetal calf serum and 1% non-essential amino acids (Invitrogen). All of the cell lines were obtained from ATCC, with the following exceptions: SNB75 (Dr D. Stojdl, Children's Hospital of Eastern Ontario Research Institute) and SF539 (UCSF Brain Tumor Bank). Primary NF1$^{-/+}$p53$^{-/+}$ cells were derived from C57Bl/6 J $p53^{+/-}/NF1^{+/-}$ mice[49]. Cell lines were regularly tested for mycoplasma contamination. BTICs were cultured in serum-free culture medium supplemented with EGF and FGF-2 (ref. 50). For siRNA transfections, cells were reverse transfected with Lipofectamine RNAiMAX (Invitrogen) for 48 h as per the manufacturer's protocol.

**Viruses.** The Indiana serotype of VSV was used in this study. VSV-EGFP, VSVΔ51 (lacking amino acid 51 in the M gene) and Maraba-MG1 were propagated in Vero cells and purified on an OptiPrep gradient. VSVΔ51 with the deletion of the gene encoding for glycoprotein (VSVΔ51ΔG) was propagated in HEK293T cells that were transfected with pMD2-G using Lipofectamine 2000 (Invitrogen), and purified on a sucrose cushion. NRRPs were generated by exposing VSV-EGFP to ultraviolet (250 mJ cm$^{-2}$) using a XL-1000 ultraviolet crosslinker (Spectrolinker)[21].

**In vitro viability assay.** Cell lines were seeded in 96-well plates and incubated overnight. Cells were treated with vehicle (0.05% DMSO) or LCL161 and infected with the indicated multiplicity of infection (MOI) of virus or treated with 1 μg ml$^{-1}$ IFN-αB/D, 0.1 ng ml$^{-1}$ TNF-α or the indicated of NRRPs for 48 h. Cell viability was determined by Alamar blue (Resazurin sodium salt (Sigma)) and data were normalized to vehicle treatment. The chosen sample size is consistent with previous reports that used similar analyses for viability assays, but no statistical methods were used to determine sample size[16,22].

**Western blotting.** Cells were scraped, collected by centrifugation and lysed in RIPA lysis buffer containing a protease inhibitor cocktail (Roche). Tumours were excised, minced and lysed as above. Equal amounts of soluble protein were separated on polyacrylamide gels followed by transfer to nitrocellulose membranes. Individual proteins were detected by western blotting using for cFLIP (7F10, 1:500, from Alexis Biochemicals) and β-tubulin (1:1,000, E7 from Developmental Studies Hybridoma Bank). Rabbit anti-rat IAP1 and IAP3 polyclonal antibodies were used to detect human and mouse cIAP1/2 and XIAP, respectively (1:5,000)[51] (Cyclex). AlexaFluor680 (Invitrogen) or IRDye800 (Li-Cor) (1:2,500) were used to detect the primary antibodies, and infrared fluorescent signals were detected using the Odyssey Infrared Imaging System (Li-Cor). Full-length blots are located in Supplementary Fig. 18.

**ELISA.** For detection of TNF-α in vivo, mice were treated with 50 μg poly(I:C) i.p. or 5 × 10$^8$ PFU of VSVΔ51 i.v. Brains were homogenized in 20 mM HEPES-KOH (pH 7.4), 150 mM NaCl, 10% glycerol and 1 mM MgCl$_2$, supplemented with EDTA-free protease inhibitor cocktail (Roche). NP-40 was added to final concentration of 0.1% and clarified through centrifugation. Equal amounts were processed for the detection of TNF-α with the TNF-α Quantikine assay kit (R&D Systems).

To assess the specificity of the adaptive immune response, mice cured of CT-2A tumours by SMC and anti-PD-1 treatment and age-matched control (naive) C57BL/6 female mice were injected subcutaneously with 1 × 10$^6$ CT-2A cells. After seven days, splenocytes were isolated and cocultured with CT-2A cells for 48 h (20:1 ratio of splenocytes to cancer cells) in the presence of vehicle or 5 μM SMC or 20 μg ml$^{-1}$ of the indicated antibodies. The secretion of IFN-γ, GrzB, TNF-α, IL-17, IL-6 and IL-10 was determined by ELISA (kits are from R&D Systems).

**CT-2A and GL261 brain tumour models.** Female 5-week-old C57BL/6 or CD-1 nude mice were anesthetized with isofluorane and the surgical site was shaved and prepared with 70% ethanol. A total of 5 × 10$^4$ cells were stereotactically injected in a 10-μl volume into the left striatum over 1 min into the following coordinates: 0.5 mm anterior, 2 mm lateral from bregma and 3.5 mm deep. The skin was closed using surgical glue. Mice were treated with either vehicle (30% 0.1 M HCl, 70% 0.1 M NaOAc pH 4.63) or 75 mg kg$^{-1}$ LCL161 orally and intratumorally (i.t.) in 10 μl with 50 μg poly(I:C), i.v. with 5 × 10$^8$ VSVΔ51 or i.p. with 250 μg of anti-CD4 (GK1.5), anti-CD8 (YTS169.4), anti-PD-1 (J43) or CTLA-4 (9H10). For treatment with birinapant, mice were treated with vehicle (12.5% Captisol) or 30 mg kg$^{-1}$ birinapant (i.p.). In some cases, animals were treated with anti-IFNAR1 (MAR1-5A3), anti-IFN-γ (R4-6A2) or anti-TNF-α (XT3.11). Isotype control IgG antibodies were used as appropriately: BE0091, BE0087, BP0090, MOPC-21 or HPRN. All neutralizing and control antibodies were from BioXCell. For intracranial cotreatment of SMC and type I IFN, mice were injected 10 μl i.t. with combinations of vehicle (0.5% DMSO), 100 μM LCL161, 0.01% BSA or 1 μg

IFN-αB/D. Alternatively, mice were treated orally with vehicle or 75 mg kg$^{-1}$ LCL161 and 1 μg IFN-α B/D (i.p.). Animals were euthanized when they showed predetermined signs of neurologic deficits (failure to ambulate, weight loss > 20% body mass, lethargy, hunched posture). Treatment groups were assigned by cages and each group had 5–9 mice for statistical measures (Kaplan–Meier with log-rank analysis). There was no randomization and the lead investigator was blinded to group allocation. The sample size is consistent with previous reports that examined tumour growth and mouse survival following cancer treatment but no statistical methods were used to determine sample size[16,52].

**MRI.** Live mouse brain MRI was performed at the University of Ottawa pre-clinical imaging core using a 7 Tesla GE/Agilent MR 901. Mice were anaesthetized for the MRI procedure using isoflurane. A 2D fast spin echo sequence (FSE) pulse sequence was used for the imaging, with the following parameters: 15 prescribed slices, slice thickness = 0.7 mm, spacing = 0 mm, field of view = 2 cm, matrix = 256 × 256, echo time = 25 ms, repetition time = 3,000 ms, echo train length = 8, bandwidth = 16 kHz, 1 average, and fat saturation. The FSE sequence was performed in both transverse and coronal planes, for a total imaging time of about 5 min.

**EMT6 mammary tumour model.** Mammary tumours were established by injecting 1 × 10$^5$ EMT6 cells in the mammary fat pad of 5-week-old female BALB/c mice. Mice with palpable tumours (∼100 mm$^3$) were cotreated with either vehicle (30% 0.1 M HCl, 70% 0.1 M NaOAc pH 4.63) or 50 mg kg$^{-1}$ LCL161 orally and either i.t. injections of 5 × 10$^8$ PFU of VSVΔ51 or i.p. injections of control IgG (BE0091) or anti-PD-1 (J43). Animals were killed when tumours metastasized i.p. or when the tumour burden exceeded 2,000 mm$^3$. Tumour volume was calculated using $(\pi)(W)^2(L)/4$, where $W$ = tumour width and $L$ = tumour length. Treatment groups were assigned by cages and each group had 4 to 5 mice for statistical measures (mean, standard error; Kaplan–Meier with log-rank analysis). There was no randomization and the lead investigator was blinded to group allocation.

**MPC-11 multiple myeloma model.** A mouse model of multiple myeloma and plasmacytoma was established by injecting 1 × 10$^6$ luciferase-tagged MPC-11 cells (i.v.) into female 4–5-week-old BALB/c mice. Mice were treated with vehicle (30% 0.1 M HCl, 70% 0.1 M NaOAc pH 4.63) or 75 mg kg$^{-1}$ LCL161 orally and with 250 μg of control IgG or α-PD-1 antibodies (i.p). Bioluminescence imaging was captured with a Xenogen 2000 IVIS CCD-camera system (Caliper Life Sciences) following i.p. injection of 4 mg luciferin (Gold Biotechnology). Treatment groups were assigned by cages and each group had 3 to 4 mice for statistical measures (Kaplan–Meier with log-rank analysis). There was no randomization and the lead investigator was blinded to group allocation.

**Tumour rechallenge.** Naive age-matched female C57BL/6 mice or mice previously cured of intracranial CT-2A tumours by SMC-based combination treatment with immunostimulants (minimum of 180 days post implantation) were reinjected with CT-2A cells i.c. as described above or with 5 × 10$^5$ cells subcutaneously. Naive BALB/c or mice previously cured of luciferase-tagged EMT6 mammary tumours with SMC and VSVΔ51 combination treatment (90–120 days post implantation) were reinjected with 5 × 10$^5$ untagged EMT6 cells in the fat pad. Animals were euthanized as described above. Blinding or randomization was not possible.

All animal experiments were conducted with the approval of the University of Ottawa Animal Care and Veterinary Service in accordance with guidelines established by the Canadian Council on Animal Care.

**Flow cytometry.** For in vitro analysis, cells were treated with vehicle (0.01% DMSO) or 5 μM LCL161 and 0.01% BSA, 1 ng ml$^{-1}$ TNF-α, 250 U ml$^{-1}$ IFN-β or 0.1 MOI of VSVΔ51 for 24 hr. Cells were released from plates with enzyme-free dissociation buffer (Gibco) and stained with Zombie Green and the indicated antibodies. For analysis of tumour immune infiltrates, intracranial CT-2A tumours were mechanically dissociated, RBCs lysed in ACK lysis buffer and stained with Zombie Green and the indicated antibodies. In some cases, cells were stimulated with 5 ng ml$^{-1}$ PMA and 500 ng ml$^{-1}$ Ionomycin in the presence of Brefeldin A for 5 h, and intracellular antigens were processed using BD Cytofix/Cytoperm kit. Antibodies include Fc Block (101319, 1:500), PD-L1 (10F.9G2, 1:250), PD-L2 (TY25, 1:100), I-A/I-E (M5/114.15.2, 1:200) and H-2K$^d$/H-2D$^d$- (34-1-2S, 1:200), CD45 (30-F11, 1:300), CD3 (17A2, 1:500), CD4 (GK1.5, 1:500), CD8 (53-6.7, 1:500), PD-1 (29.1A12, 1:200), CD25 (PC61, 1:150), Gr1 (RB6-AC5, 1:200), F4/80 (BM8, 1:200), GrzB (GB11, 1:150) and IFN-γ (XMG1.2, 1:200). All antibodies were from BioLegend except for TNF-α (MP6-XT22, 1:200) and CD11b (M1/70, 1:100) where from BD Biosciences. Cells were analysed on a Cyan ADP 9 (Beckman Coulter) or BD Fortessa (BD Biosciences) and data were analysed with FlowJo (Tree Star).

**Microscopy.** Detection of mKate2-CT-2A cells was performed in an incubator outfitted with an Incucyte Zoom microscope equipped with a × 10 objective. Enumeration of fluorescent signals from the Incucyte Zoom was processed using the integrated object counting algorithm within the Incucyte Zoom software.

**Multiplex ELISA.** The detection of serum proteins following combinatorial SMC and anti-PD-1 treatment was analysed by a flow cytometry-based multiplex kit (LEGENDplex inflammation panel from Biolegend). Hierarchical analysis was determined using Morpheus (https://software.broadinstitute.org/morpheus).

**RT-qPCR.** Total RNA from mice was isolated from mice treated with combinations of LCL161 and anti-PD-1 using the RNAEasy Mini Plus Kit. Two step RT-qPCR was performed using iScript and SsoAdvanced SYBR Green supermix (BioRad) on a Mastercycler ep realplex (Eppendorf). The library of cytokine and chemokine genes was from realtimeprimers.com. A $n = 4$ was performed for each treatment conditioned and data were normalized to eight different reference genes and compared with each vehicle and IgG sample. The data were analysed by hierarchical analysis using Morpheus.

**ELISpot.** CD8$^+$ T-cells were enriched from splenocytes of female age-matched naive mice or mice previously cured of intracranial CT-2A (180 days post-implantation) or mammary EMT6 tumours (120 days post-implantation) using a CD8 magnetic selection kit (Stemcell Technologies). CD8$^+$ cells were cocultured with cancer cells (1:20 for CT-2A, LLC, and 1:12.5 for EMT6 or 4T1 cells) and with 10 μg ml$^{-1}$ IgG (BE0091) or anti-PD-1 (J43) for 48 h using the IFN-γ or Granzyme B ELISpot kits (R&D Systems).

**Statistical analysis.** Comparison of Kaplan–Meier survival plots was conducted by log-rank analysis and subsequent pairwise multiple comparisons were performed using the Holm–Sidak method (SigmaPlot). Comparison between multiple treatment groups was analysed using one-way ANOVA followed by *post hoc* analysis using Dunnett's multiple comparison test with adjustments for multiple comparison (GraphPad). Estimate of variation was analysed with GraphPad. Comparison of treatment pairs was analysed by two-sided *t*-tests (GraphPad).

**Data availability.** The data that support the findings of this study are available from the corresponding authors on reasonable request.

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

## Acknowledgements

We are grateful to Dr David P. Conrad (Celverum) and Drs John C. Bell, Cory Batenchuk and Fabrice Le Boeuf (OHRI) for providing live and ultraviolet-inactivated viruses. We appreciate the gift of the SMCs LCL161 from Novartis and Birinapant from Tetralogic Pharmaceuticals. We thank Drs Gregory Cron (University of Ottawa) and Lawton Stubbert (OHRI) for their help in establishing and visualizing intracranial mouse models of cancer. We thank Drs Artee Luchman and Sam Weiss for providing patient BTIC lines (University of Calgary). This work was supported by an Impact grant co-funded by the Canadian Cancer Society Research Institute (CCSRI) and Brain Canada (#704119; R.G.K., T.A., S.T.B. and E.C.L.) and operating grants from the Canadian Institutes of Health Research (CIHR, #231421, #318176, #361847; R.G.K., S.T.B. and E.C.L.) and by funding from the Ottawa Regional Cancer Foundation (ORCF)/Ottawa Kiwanis Medical Foundation, and the Children's Hospital of Eastern Ontario (CHEO)/ Neuroblastoma Foundation; R.G.K. and E.C.L.

## Author contributions

S.T.B., E.C.L., T.A. and R.G.K. conceived and directed the project. S.T.B. and E.C.L. wrote the manuscript. T.A. and R.G.K. edited the manuscript. S.T.B., C.E.B., C.H., T.S., M.S.-J., D.E.W., J.C., N.E., A.M. and X.Q.L. performed and analysed experiments. D.L.S., S.M.R., P.S. and P.A.F. provided key services, materials and guidance and commented on manuscript.

## Additional information

**Competing financial interests:** The authors declare no competing financial interests.

**Reprints and permission** information is available online at http://npg.nature.com/ reprintsandpermissions/

**DOI: 10.1038/ncomms16231**       **OPEN**

# Publisher Correction: Smac mimetics synergize with immune checkpoint inhibitors to promote tumour immunity against glioblastoma

Shawn T. Beug, Caroline E. Beauregard, Cristin Healy, Tarun Sanda, Martine St-Jean, Janelle Chabot, Danielle E. Walker, Aditya Mohan, Nathalie Earl, Xueqing Lun, Donna L. Senger, Stephen M. Robbins, Peter Staeheli, Peter A. Forsyth, Tommy Alain, Eric C. LaCasse & Robert G. Korneluk

*Nature Communications* 8:14278 doi: 10.1038/ncomms14278 (2017); Published 15 Feb 2017, Updated 18 Jul 2018

The original HTML version of this Article omitted the article number; it should have been '14278'. This has now been corrected in the HTML version of the Article. The PDF version was correct from the time of publication.

