## [Peer Review File · Nature Communications]

Reviewers' comments:

Reviewer #1 (Remarks to the Author):

This manuscript presents relevant evidence for the combinations of drugs enhancing cIAPs degradation and immunotherapy/virotherapy. For this purpose the LCL161 compound of Novartis is thoroughly tested. It would be very reassuring if another smac mimetic compound would be tested in key experiments to rule out immune off target effects of the Novartis drug. Following evidence for synthetic lethality on cultures of glioma cell lines, combination treatment experiments are performed with VSG-based virotherapy, systemic and intracranial Poly I:C, type I IFN as well as with anti-CTLA-4 and antiPD-1 check point inhibitors. The models are chiefly transplantable gliomas in mice but some experiments have been undertaken with other mouse transplantable tumors. Combinations of immunotherapy with smac-mimetics are ongoing in clinical trials. The theme is therefore of interest and high grade gliomas are currently in the limelight of immunotherapy.

Comments:

1. As mentioned more than one compound would be reassuring with regard to off target effects.
2. Is IFN α /beta a common key player in the therapeutic effects? Could IFNAR be blocked with antibody to check for the therapeutic outcome.
3. Is the myeloid compartment in intracranial tumors modified by LCL161 treatment. Is the pattern of genes expressed by such cells modified by cIAP downmodulation (as suggested in figure 6 as a mechanism of action).
4. The IFN β /D chimeric molecule should be detailed. Is it needed or any type I IFN can do the same job?
5. Is there evidence for activation of the alternative NF- κ B pathway in the tumors undergoing rejection, particularly in the T cell compartment?
6. What would anti-PD-1 +anti-CTLA-4 + LCL161 achieve? ...Triple synergy leading to cure in every case. This could be relevant due to the very active clinical development of these combinations with immunomodulatory monoclonal antibodies.
7. Figure 6 is highly speculative and many facts are not well justified by the data. The level of certainty should be disclosed and the proper backing each step references should be included in the figure itself, since it is mostly from papers published elsewhere and some of them controversial.

Reviewer #2 (Remarks to the Author):

Smac mimetics are molecules which mimic the inhibitory activity of the endogenous molecule Smac in inhibiting the function of IAP (inhibitor-of-apoptosis) proteins. These molecules have been developed as targeted agents to sensitize cancer cells to the extrinsic cell death pathway by overcoming (X)IAP-conferred resistance. The present study reports that Smac mimetics can synergize with immunotherapy in controlling tumor growth in mouse glioma models, in a cytotoxic CD8 T cell-dependent manner.

Comments

1 Figure 1. Lacks statistics and I would not call this "immunostimulants" in the legend if in fact this was an "immune system" free in vitro design.

2 Figure 2 and text: unless the authors demonstrate that their CT-2A tumors are protected by a blood brain barrier, they must avoid speculating whether or not SMCs passed the barrier in their experiments. Preferentially they should do similar experiments in tumor-free healthy mice and look

at endogenous brain IAP.

3 Figure 2: Systemic IFN-alpha plus systemic SMC would be more relevant to the clinical setting. Local treatment in the mice can hardly be extrapolated to the complex configuration of the human disease where essentially all local therapy approaches have (more or less) failed.

4 Do the mouse glioma models express PD-L1 in vivo?

5 Were any histological studies done to understand why there was synergy between Smac mimetics and immune checkpoint blockade?

6 The concept of using Smac mimetics for the treatment is not entirely novel, but was in fact described for the 10 years ago, with a focus on sensitization to Apo2L - this should be discussed (Fulda et al. Nat Med. 2002 Aug;8(8):808-15).

7 The source of the PD-1 and CTLA-4 antibodies should be indicated.

Reviewer #3 (Remarks to the Author):

This study by Beug et al. shows that Smac mimetic SMC LC161 synergizes with innate immunostimulants (oncolytic virus) and immune checkpoint inhibitors (anti-PD-1 or CTLA4 antibodies) to produce durable cures in mouse models of glioblastoma. This study represents a very nice extension of their previous studies showing that SMCs synergize with innate immune stimuli to stimulate a potent but safe "cytokine storm" to kill tumor cells (Beug, et al. Nature Biotechnology, 2014). Clinical trials of Smac mimetics including LC161 have advanced to Phase 2 and further development of this class of compounds offers therapeutic potentials. Several different immunotherapy agents have been approved. Thus, this study has immediate translational values.

The combinations of SMC LC161 with innate immunostimulants (oncolytic virus) and immune checkpoint inhibitors (anti-PD-1 or CTLA4 antibodies) significantly extended the survival of tumor bearing mice, which are quite convincing and exciting. Additional mechanistic data are needed to strengthen the claims. Specifically:

1. The authors concluded that IAP antagonists incorporate both innate and adaptive immunity to cure mice of cancer (abstract). While CD8+ T cells were shown to be required for combination efficacy (Fig 5b), the lymphocyte and leukocyte infiltration to the tumors has not been explicitly examined. In addition to CD8+ T cells, what other immune cells are altered upon combination treatment in tumors?

2. Fig. 5a shows a significant increase in the production of IFN γ and granzyme B from CT-2A cells co-incubated with splenocytes from surviving mice. What's more relevant is: is there any difference/change in the levels of intratumoral cytokines and chemokines with mono- and SMC LC161 combination therapies? What's the effect of CD8+ T cells depletion on the profiles of intratumoral cytokines and chemokines?

3. CD8+ T cells produce IFN γ during anti-tumor responses. It's suggested by the authors that IFN and/or cytokine response is responsible for cell killing in the SMC combination settings. Thus, it will be very informative to know the contribution of IFN γ in the activities of SMC combinations. Will anti-IFN γ antibody reduce the therapeutic efficacy of SMC combination?

General comments

To address the reviewer's specific comments outlined below, we included new data, which necessitated the generation of 4 completely new figures (Figs. 6-9) in the main text (and 7 new figures in supplemental), as well as new figure panels (main and supplementary), re-arrangement of some figures and minor modification of one figure. The changes to the figures are outlined below:

Figure 1 – no change

Figure 2 – (a) inclusion of additional Western blots; (g) inclusion of new survival data

Figure 3 – no change

Figure 4 – (b, c, f, g) new data; (d, e) were formerly Figure 4b and c

Figure 5 – (a) re-interpretation of the existing data as heat maps. Box and whisker plots were moved to Supplementary Figure 13a; (b, c) new data; (d, e) were formerly Figure 5b and c

Figure 6 – new data

Figure 7 – new data

Figure 8 – new data

Figure 9 – new model

Supplementary Figure 1 – new data

Supplementary Figure 2 – formerly Supplementary Figure 1

Supplementary Figure 3 – formerly Supplementary Figure 2

Supplementary Figure 4 – formerly Supplementary Figure 3

Supplementary Figure 5 – new data (relates to Figure 2a)

Supplementary Figure 6 – formerly Supplementary Figure 4

Supplementary Figure 7 – formerly Supplementary Figure 5

Supplementary Figure 8 – formerly Supplementary Figure 6

Supplementary Figure 9 – formerly Supplementary Figure 7

Supplementary Figure 10 – new data (relates to Fig. 4b)

Supplementary Figure 11 – formerly Supplementary Figure 8

Supplementary Figure 12 – formerly Supplementary Figure 9

Supplementary Figure 13 – new data (relates to Fig. 5a, b)

Supplementary Figure 14 – new data (relates to Fig. 7e)

Supplementary Figure 15 – new data (relates to Fig. 7f)

Supplementary Figure 16 – new data

Supplementary Figure 17 – formerly Figure 6, (minor modifications to address reviewer concerns)

Collectively, our new data relates to:

1. Penetrance of SMCs across the BBB penetrance: Fig. 2a; Supplementary Fig. 5
2. Systematic treatment with IFN α and SMC: Fig. 2g
3. Assessment of other SMCs: Fig. 4f, 5c; Supplementary Fig. 1
4. Immunophenotyping and roles of T-cells: Fig. 4b, 6, 7a,b; Supplementary Fig. 10, 16
5. Expression of PD-L1 in vivo: Fig. 4c
6. Expression of cytokines and chemokines in serum and in the tumor microenvironment: Fig. 5b, 7c-f, 8a; Supplementary Fig. 13-15
7. Assessment of anti-PD-1, anti-CTLA4 and SMC: Fig. 4g
8. Functional roles of cytokines: 7g, 8b, 8c

Point-by-point response

Reviewer #1 (Remarks to the Author): The referee states clearly that they are convinced of the synergy in killing brain tumor cells. Moreover, the referee emphasizes that “*The theme is therefore of interest and high grade gliomas are currently in the limelight of immunotherapy*”. The main criticisms are that “*another smac mimetic compound would be tested in key experiments to rule out immune off target effects of the Novartis drug*” and to further elucidate the mechanisms involving Smac mimetic efficacy.

Remark 1. As mentioned more than one compound would be reassuring with regard to off target effects.

Response: We have now tested other SMCs. While there are several different SMCs being developed by companies and academic groups, there are two main structural classes of SMCs: monovalent and bivalent, both which potently and selectively target cIAP1, cIAP2 and XIAP. Monovalent SMCs consist of a single SMC compound while bivalent SMCs are two SMCs tethered by a chemical linker. Monomer SMCs, such as LCL161, are administered to patients orally while bivalent SMCs, such as birinapant, are injected intravenously. Due to the ability of bivalent SMCs to target two different BIR domains of the IAPs, bivalent SMCs are approximately 10-100 fold more potent than monomer SMCs in vitro. We now report that the different SMCs are mechanistically equivalent in the ability to kill cancer cells in the presence of TNF- α (Supplementary Fig. 1). We also observe that the combination of anti-PD-1 and birinapant was able to significantly induce the regression of intracranial CT-2A tumors (Fig. 4f). Lastly, in a coculture system employing CT-2A cells and CD8 T-cells derived from mice previously cured of intracranial CT-2A cells, we observed pronounced death of CT-2A cells only in the presence of SMCs (Fig. 5c).

Remark 2. Is IFNalpha/beta a common key player in the therapeutic effects? Could IFNAR be blocked with antibody to check for the therapeutic outcome.

Response: We agree that this an interesting issue and have done the experiment. Initially we assessed if the level of type I IFN is changed with respect to combination treatment of SMC and ICI. We observed that, in doubly treated mice, the type I IFN IFN- β protein levels are upregulated in the serum (Fig. 7e; Supplementary Fig. 14) and there is a general increase of RNA levels of IFN- β within brain tumors (Fig. 7f; Supplementary Fig. 15). To functionally assess the requirement for type I IFN signaling in mice, we administered an antibody that blocks type I IFN signaling in mice that were co-treated with SMC and anti-PD-1 antibodies. This experiment revealed that type I IFN signaling is required in part for the synergy between SMCs and ICIs in eradicating glioblastoma tumors (Fig. 7g).

Remark 3. Is the myeloid compartment in intracranial tumors modified by LCL161 treatment? Is the pattern of genes expressed by such cells modified by cIAP downmodulation (as suggested in figure 6 as a mechanism of action).

Response: To assess whether specific myeloid cells were affected by SMC treatment (and in combination with anti-PD-1 antibodies), we performed flow cytometry to detect the proportion of monocyte derived suppressor cells (MDSCs) and macrophages/microglia from mice bearing intracranial CT-2A tumors. We did not observe significant differences in the proportion of MDSCs (CD11b⁺ Gr1⁺) among the treatment cohorts (Fig. 6f). However, there was a general significant trend of reduced

presence of macrophages/microglia in these tumors (Fig. 6g), which is attributed to the effects of the anti-PD-1 antibody.

As our focus in this manuscript relates to the effects of IAP antagonism on cytotoxic T-cells towards inducing glioblastoma cell death, we did not profile gene expression of these cells in response to SMC treatment. In addition, the mechanistic role of IAP antagonism in macrophages towards the eradication of mouse mammary tumors is currently a focus of a different study.

Remark 4. The IFN β /D chimeric molecule should be detailed. Is it needed or any type I IFN can do the same job?

Response: To provide more context regarding IFN α B/D, we incorporated a statement that explains the hybrid IFN- α molecule: Page 7, “For in vivo studies, we used a form of recombinant IFN- α that consists of a hybrid of human isoforms IFN- α B and IFN- α D, which displays potent antiviral activity among a broad range of species²⁴.” We have previously evaluated the ability of recombinant IFN α B/D or murine IFN β to eradicate various murine tumors. We observed strong synergy with the combination of SMC and IFN α B/D but did not observe significant synergy with SMC and IFN β co-treatment (unpublished data). Hence, we used the IFN α B/D for the current study. To date, we have not tried recombinant mouse IFN- α , and we do not know the reason why we fail to observe synergy between SMC and IFN- β .

Remark 5. Is there evidence for activation of the alternative NF- κ B pathway in the tumors undergoing rejection, particularly in the T cell compartment?

Response: There are a plethora of reports that document that SMC treatment, by virtue of downregulating cIAP1 and cIAP2, engages the alternative NF- κ B signaling pathway. While we have characterized the role of the alternative NF- κ B in inducing death of cancer cells in the presence of TNF- α , through pharmacological and genetic means, we do not possess the repertoire of transgenic mice and array of small molecule inhibitors to address this concern. This is nevertheless an interesting question, as the SMC-mediated effects are highly multimodal in immune cells, involving lymphoid and myeloid cells, and other host cells. Nevertheless, we show that SMC treatment enhances the production of TNF- α in splenic cells and in isolated CD8 T-cells (Fig. 5a, b; Fig. 7d-f; Fig. 8a) which is in agreement with SMC enhancing proinflammatory responses upon stimulation with an appropriate trigger.

Remark 6. What would anti-PD-1 + anti-CTLA-4 + LCL161 achieve? ...Triple synergy leading to cure in every case. This could be relevant due to the very active clinical development of these combinations with immunomodulatory monoclonal antibodies.

Response: We agree that it is an interesting prospect whether the triple combination of these two ICIs and a SMC would lead to a greater cure rate in the CT-2A brain tumor model, and we accordingly conducted this experiment. We report that that the combination of anti-PD-1 and anti-CTLA4 results in a 66% survival rate, and the inclusion of SMC in this double treatment cohort results in a 100% survival rate (Fig. 4g).

Remark 7. Figure 6 is highly speculative and many facts are not well justified by the data. The level of certainty should be disclosed and the proper backing each step references should be included in the figure itself, since it is mostly from papers published elsewhere and some of them controversial.

Response: We agree with the reviewer that certain aspects of this model are speculative and are not directly supported by the current study. This schematic, nevertheless, is a unifying concept that bridges our data and other reports for the contribution of innate and adaptive immunity to eradicate cancer. We wished to retain this message and therefore moved this figure to Supplementary Fig. 17. We also slightly modified the model to emphasize our new findings, as well as those which are well established. For example, we show that SMCs can induce the production of TNF- α from various immune cells, using macrophages and T-cells as an example.

In its place, we included a new model that focuses specifically on the main findings of this study that addresses mechanism of SMC and anti-PD-1 synergy (Fig. 9).

Reviewer #2 (Remarks to the Author): Consistent with Reviewers #1 and #3, the reviewer recommends additional experiments to further understand the mechanisms, and to probe for penetrative ability of Smac mimetics across the BBB.

Remark 1. Figure 1. Lacks statistics and I would not call this "immunostimulants" in the legend if in fact this was an "immune system" free in vitro design.

Response: We have rephrased the terminology used for the agents, and now state that we used cytokines and oncolytic viruses in the legend of Figure 1. As requested, instead of performing statistical analysis for experiments using animal studies, we also we conducted statistical analysis for all in vitro work. Due to the nature of high reproducibility of in vitro work, we observe statistical significance when there is more than a ~5% change in cell viability. As we believe that this type of response is not biologically meaningful in inducing robust death of cancer cells, we only report in vitro statistical significance if the p level is < 0.0001 (Fig. 1, 2e; Supplementary Fig. 1, 2, 3, 11a, 12a). We have added comments in the figure legends which explain the statistical methodology and analysis that was performed.

Remark 2. Figure 2 and text: unless the authors demonstrate that their CT-2A tumors are protected by a blood brain barrier, they must avoid speculating whether or not SMCs passed the barrier in their experiments. Preferentially they should do similar experiments in tumor-free healthy mice and look at endogenous brain IAP.

Response: We agree and have now included this data. When we initially extracted CT-2A brain tumors after SMC treatment for Western blotting, we also extracted the remaining brain matter from these mice to detect the levels of the SMC targets, cIAP1, cIAP2 and XIAP. We now report that in these mice that had CT-2A tumors, the level of cIAP1 was not similarly downregulated in adjacent non-tumor brain matter (Fig. 2a). Furthermore, as measured by mass spectrometry, different groups and companies with different versions of SMCs have not detected SMC or its metabolites in the brain in intact animal species. Nevertheless, we assessed whether SMC are able to cross the BBB and induce degradation of the IAPs in non-tumor bearing mice. We now report that oral gavage of 75mg/kg LCL161 does not induce the degradation of the IAPs within brain tissues in non-tumor bearing mice (Supplementary Fig. 5). In this experiment, the spleen served as a positive control for IAP degradation as mediated by SMC treatment. We conclude that SMC can reach tumor tissue in mice likely through a compromised BBB.

Remark 3. Figure 2: Systemic IFN-alpha plus systemic SMC would be more relevant to the clinical setting. Local treatment in the mice can hardly be extrapolated to the complex configuration of the human disease where essentially all local therapy approaches have (more or less) failed.

Response: We initially restricted the delivery of IFN- α B/D intracranially as it has been reported that this cytokine does not induce the expression of its gene targets in the brain. However, we were sufficiently intrigued to determine whether systemic delivery of IFN- α along with oral gavage of SMC would be efficacious in inducing regression of glioblastoma in mice bearing intracranial CT-2A tumors. We now report that the dual systemic administration of recombinant IFN- α B/D with SMC in mice bearing intracranial CT-2A tumors leads to durable cures in over half of the double treated mice (Fig. 2g). Although it remains to be mechanistically determined, these results imply that the synergistic effects of systemic administration of SMC and VSV Δ 51 (Fig. 2c) may be as a result of the global induction of type I IFN, which then leads to the presence of TNF- α within the brain (Fig. 2a, 7e-g).

Remark 4. Do the mouse glioma models express PD-L1 in vivo?

Response: In agreement with the in vitro data (Fig. 3a), we show that intracranial CT-2A cells treated with vehicle or SMC express significant levels of PD-L1 (Fig. 4c). In this experiment, we excised CT-2A brain tumors following SMC treatment, and analyzed the expression level of PD-L1 in the CD45 negative population, which are predominantly CT-2A cells. However, SMC treatment does not further augment PD-L1 expression.

Remark 5. Were any histological studies done to understand why there was synergy between Smac mimetics and immune checkpoint blockade?

Response: We initially attempted to conduct histological studies to assess for the infiltration of immune cells and to examine for death of tumor cells. However, our initial studies were not successful, as we treated mice at 14 d post-implantation with 4 anti-PD-1 and 3 SMC treatments, and we only observed one tumor out of 6 mice in that treatment cohort. We then attempted treatment at a later post-implantation time point and included less treatments (Fig. 6a). At this point, the resulting tumors from the doubly treated mice were small and we decided to process the tumors for analysis by flow cytometry. We report that SMC treatment:

- results in an increase of CD8 T-cells positive for PD-1 (Fig. 4c)
- reduces CD4 and CD25 positive population upon SMC treatment (Fig. 6c), indicative of a less pronounced population of immunosuppressive T-cells
- decreases PD-1 positive CD4 T-cells in the double treatment cohort (Fig. 6e)
- reduces the proportion of macrophages/microglia in ICI or SMC and ICI treated mice (Fig. 6g)
- increases Granzyme B positive CD8 T-cells in anti-PD-1 or anti-PD-1 and SMC treated mice (Fig. 7a)
- decreases the production of IFN- γ in SMC-treated CD4 T-cells (Fig. 7c)
- upregulates TNF- α expression in T-cells upon SMC treatment in vivo (Fig. 7d) and ex vivo (Fig. 8a).

Remark 6. The concept of using Smac mimetics for the treatment is not entirely novel, but was in fact described for the 10 years ago, with a focus on sensitization to Apo2L - this should be discussed (Fulda et al. Nat Med. 2002 Aug;8(8):808-15).

Response: This previous study (which is now cited as Ref. 40 on Page 17 in the discussion) examined the effect of antagonizing the IAPs using cell-permeable Smac peptides and recombinant TRAIL in an immunodeficient xenograft model of glioblastoma. In contrast, we are using an immunocompetent mouse with a clinical candidate small molecule that mimics the function of active Smac, and are relying on the systemic and local induction of cytokines with SMC treatment.

Remark 7. The source of the PD-1 and CTLA-4 antibodies should be indicated.

Response: All antibodies used for in vivo experiments were obtained from BioXCell (Pages 22 and 23). We also listed the clonality and the isotype IgG controls in the revised manuscript.

Reviewer #3 (Remarks to the Author): The referee agrees that this study has “*has immediate translational values*” and suggests that “*Additional mechanistic data are needed to strengthen the claims*”.

Remark 1. The authors concluded that IAP antagonists incorporate both innate and adaptive immunity to cure mice of cancer (abstract). While CD8+ T cells were shown to be required for combination efficacy (Fig 5b), the lymphocyte and leukocyte infiltration to the tumors has not been explicitly examined. In addition to CD8+ T cells, what other immune cells are altered upon combination treatment in tumors?

Response: The analysis of tumor infiltrating immune cells were similarly requested by Reviewer #1 (*Remark 3*) and Reviewer #2 (*Remark 5*), and the response to this remark has been covered. To summarize, in mice bearing intracranial CT-2A tumors and treated with combinations of SMC and anti-PD-1, we examined for the expression level of PD-1, GrzB, TNF- α in T-cells within tumors and for the levels of infiltrating MDSCs and macrophages/microglia (Fig. 6, 7a-d, 8a). We restricted our in vivo analysis to these cell types and molecules as they are pertinent to the ability of SMCs to elicit death of cancer cells by adaptive immune cells, such as CTLs, through GrzB and TNF- α production. We have previously established that SMCs and immunostimulants make use of the innate immune system to cure mice of solid tumors through the induction of TNF- α (Beug et al Nature Biotech 2014). In this study, we demonstrate that SMCs when combined with ICIs make use of the adaptive immune system as well to kill cancer cells. Hence SMCs are capable of using both the innate and adaptive immune system to kill cancer cells (as illustrated in the model figures, Fig. 9 and Supplementary Fig. 17).

Remark 2. Fig. 5a shows a significant increase in the production of IFN γ and granzyme B from CT-2A cells co-incubated with splenocytes from surviving mice. What's more relevant is: is there any difference/change in the levels of intratumoral cytokines and chemokines with mono- and SMC LC161 combination therapies? What's the effect of CD8+ T cells depletion on the profiles of intratumoral cytokines and chemokines?

Response: We agree that this analysis is more relevant to understanding the mechanistic basis of synergy between SMC and ICI treatment. Accordingly, we first analyzed whether the cytokine response from splenic co-culture is modulated in the presence of SMC (Fig. 5b). We observed an increase in the production of pro-inflammatory factors and GrzB. Next, as suggested by the reviewer, we examined the expression profiles of genes and chemokines of CT-2A intracranial tumors in mice treated with combinations of SMC and anti-PD-1. Remarkably, we saw clustering of an increase of several factors:

the proinflammatory cytokines IFN- β , IL-1 β , IL-17, Osm and TNF- α , the chemokines Ccl2 (MCP-1), Ccl5, Ccl7, Ccl22, Cxcl9, Cxcl10 and Cxcl11, and multifaceted factors such as IL-2, IL-12 and IFN- γ . The signature expression profile of these cytokines and chemokine was not restricted to the tumor; we observed a general trend in an increase of chemokines and proinflammatory cytokines from the serum (Fig. 7e).

We have depleted CD8 T-cells in our tumor models, and we observed a loss of efficacy in doubly treated mice (Fig. 5d). However, we did not analyze the gene expression profiles of the tumors from these mice. Instead, we reasoned that, since we observed there is a consistent increase of TNF- α in our combination experiments (Fig. 5a,b; 7d-f, 8a), TNF- α is required for efficacy of CTLs against cancer cells. Therefore, we performed additional experiments to neutralize TNF- α signaling in co-culture assays and in an animal model of glioblastoma. We now report that T-cells can eradicate glioblastoma cells in SMC treated mice through the production of TNF- α globally and locally (Fig. 8b, c). This provides a compelling explanation as to why synergy is seen between SMCs and ICIs, which are two functionally and structurally disparate classes of drugs (as depicted in the model in Fig. 9). In addition, consistent with previous other reports demonstrating that the IAP antagonism enhances T-cell proliferation, we observed a significant accumulation of CFSE-negative CD8 T-cells cells after 4 days of co-culture with CT-2A in the presence of SMC (Supplementary Fig 16).

Remark 3. CD8+ T cells produce IFN γ during anti-tumor responses. It's suggested by the authors that IFN and/or cytokine response is responsible for cell killing in the SMC combination settings. Thus, it will be very informative to know the contribution of IFN γ in the activities of SMC combinations. Will anti-IFN γ antibody reduce the therapeutic efficacy of SMC combination?

Response: Indeed, we observed an increase of IFN- γ levels in the serum of doubly treated mice and an increase of IFN- γ gene expression within intracranial CT-2A tumors (Fig. 7e, f). In accordance with the reviewer's comment, we examined for the contribution of IFN- γ to the therapeutic efficacy of double treatment. We report that the inclusion of anti-IFN- γ neutralizing antibodies significantly reduced the efficacy of SMC and anti-PD-1 treatment for glioblastoma (Fig. 7g). Hence, the efficacy of SMC combination is at least partially dependent on IFN- γ signaling.

REVIEWERS' COMMENTS:

Reviewer #1 (Remarks to the Author):

The revised version is satisfactory and addressed very well my comments and those made by the other reviewers.

Reviewer #2 (Remarks to the Author):

None

Reviewer #3 (Remarks to the Author):

The authors have done an excellent job to address those points raised by all the reviewers. This manuscript is ready to be accepted.